# A widely distributed family of eukaryotic and bacterial deubiquitinases related to herpesviral large tegument proteins

Ilka Erven [1], Elena Abraham [2], Thomas Hermanns [1], Ulrich Baumann [2] & Kay Hofmann [1] ✉

Distinct families of eukaryotic deubiquitinases (DUBs) are regulators of ubiquitin signaling. Here, we report on the presence of an additional DUB class broadly distributed in eukaryotes and several bacteria. The only described members of this family are the large tegument proteins of herpesviruses, which are attached to the outside of the viral capsid. By using a bioinformatics screen, we have identified distant homologs of this VTD (Viral tegument-like DUB) family in vertebrate transposons, fungi, insects, nematodes, cnidaria, protists and bacteria. While some VTD activities resemble viral tegument DUBs in that they favor K48-linked ubiquitin chains, other members are highly specific for K6- or K63-linked ubiquitin chains. The crystal structures of K48- and K6-specific members reveal considerable differences in ubiquitin recognition. The VTD family likely evolved from non-DUB proteases and spread through transposons, many of which became 'domesticated', giving rise to the *Drosophila* male sterile *(3)76Ca* gene and several nematode genes with male-specific expression.

Ubiquitination, the posttranslational modifications of proteins by covalent attachment of ubiquitin, is an important cellular regulatory mechanism. Unlike protein phosphorylation, the modification by ubiquitin rarely activates or inactivates proteins directly, but rather regulates other aspects such as protein degradation, intracellular transport, and recruitment to particular sites[1]. An important contributor to the diversity of ubiquitin-based signals is the ability of multiple ubiquitin residues to be subject to further ubiquitination, leading to the formation of ubiquitin chains of different linkage types[2,3]. While ubiquitin chains linked through lysine residues K48 and K11 often lead to proteasomal degradation of the modified substrates, K63-linked ubiquitin chains rather affect protein localization events. Besides the abundant K48, K63 and K11-linked chains, several minor linkage types are currently attracting attention due to their highly-regulated occurrence under cellular stress situations[4–6]. Of particular interest are K6-linked chains, which have been first observed after DNA damage, where they are probably generated by the ubiquitin ligase

BRCA1[7]. K6 chains also play a role in the regulation of mitophagy, the removal of defective mitochondria by selective autophagy. Parkin, the mitophagy-associated ubiquitin ligase, can generate K6 chains and is itself modified and regulated by K6 ubiquitin[8,9]. Furthermore, K6 chains generated by the ubiquitin ligase LRSAM1 are important for the defense against intracellular bacteria via ubiquitin-dependent autophagy[10].

Deubiquitinating enzymes (DUBs) cleave peptide- and isopeptide-bonds downstream of ubiquitin. While some DUBs are required for ubiquitin activation and recycling, most DUB enzymes regulate ubiquitin signaling by undoing the activity of ubiquitin ligases - either by cleaving within ubiquitin chains or by completely removing ubiquitin from the substrate. Typical eukaryotes harbor many DUB types with different specificities for the substrates and/or the linkage of the attached ubiquitin chains[11]. Apart from a few metalloproteases belonging to the JAMM/MPN + family, most deubiquitinating enzymes are cysteine proteases, which – despite a high degree of structural

[1]Institute for Genetics, University of Cologne, Zülpicher Straße 47a, D-50674 Cologne, Germany. [2]Institute of Biochemistry, University of Cologne, Zülpicher Straße 47, D-50674 Cologne, Germany. ✉e-mail: kay.hofmann@uni-koeln.de

variability – all assume a papain-like fold[12]. In accordance with sequence relationship and structural criteria, eukaryotic cysteine-DUBs are usually categorized in six different classes[11,12]: USP (ubiquitin-specific proteases), UCH (ubiquitin carboxyl-terminal hydrolases), OTU (ovarian tumor domain proteins), Josephins (Ataxin-3–like proteins), MINDY (MIU-containing new DUB family), and ZUP1/ZUFSP (zinc-finger ubiquitin protease 1). Among those, the highly-abundant USP class comprises mostly substrate-specific DUBs, while members of the OTU class are known for their pronounced linkage specificity[13].

Many bacteria and viruses also code for deubiquitinating enzymes, which work within the host cell to counteract ubiquitin-based cellular defense responses. Most bacterial DUBs either belong to the OTU class or are members of the 'CE-clan', a class of papain-fold enzymes that includes proteases for several ubiquitin-related modifiers such as NEDD8 or SUMO[14]. Most DUB effectors secreted by intracellular bacteria prefer K63-linked chains or have no strong linkage preference; it is assumed that these enzymes remove ubiquitin from the bacterial surface or from the cytoplasmic face of bacteria-containing vacuoles, thereby helping the bacteria to evade clearance by ubiquitin-mediated targeting to the lysosome. For *Legionella pneumophila*, two exceptional DUB activities have been identified: The effector RavD specifically cleaves linear (M1-linked) ubiquitin chains, thereby inhibiting inflammation and preventing M1-based xenophagy signals. The *Legionella* effector LotA, which belongs to the OTU class, has a K6-specific DUB activity of unknown biological significance[15]. Viral deubiquitinases are typically divergent members of the USP, OTU or CE-clan families and often fulfill multiple proteolytic functions[16]. Besides removing ubiquitin, many viral DUBs can also cleave the ubiquitin-related modifier ISG15, an important interferon-stimulated component of the antiviral response[17,18]. In addition, viral DUBs are often tasked with processing viral polyproteins at multiple sites. This latter cleavage does not involve the hydrolysis of isopeptide bonds – a hallmark of DUB activity – but the cleaved peptide bonds are typically flanked by a di-glycine motif resembling the C-terminus of ubiquitin. Despite the structural differences between the cysteine DUB classes found in eukaryotes, their viruses, and prokaryotic pathogens, these enzymes not only share a common papain fold and a conserved active site topology; there are also detectable sequence similarities between different DUB classes, which have been used for predicting novel and unusual DUB families[19,20].

The protease domain found in the large tegument protein of α, β and γ herpesviruses forms a small DUB family[21–23], which differs in several respects from other deubiquitinase classes: While these proteases share the papain-like fold and the ability to cleave ubiquitin chains of various linkage types, the unusual Cys-Asp-His active site has a different architecture[24]. Moreover, extensive bioinformatical analyses did not detect any sequence relationship to other deubiquitinase classes, suggesting a different evolutionary origin[19]. A number of biological roles for the tegument-associated deubiquitinases in different herpesviruses have been proposed, including the inhibition of NF-κB and interferon signaling, and the prevention of selective autophagy[23,25–27]. BPLF1, the tegument-associated DUB from Epstein-Barr virus, was shown to also possess deneddylase activity and thereby to modulate host-encoded cullin-based ubiquitin ligases to the virus advantage[28]. The importance of the DUB domain of herpesvirus large tegument proteins for virion stability and virus entry could be separated from contributions of the unstructured C-terminal portion of the protein[29], which is required for attaching the protein to the outside of the viral capsid[30].

Based on the unusual active site architecture and the general dissimilarity to other deubiquitinases, the authors of the M48 crystal structure suggested that this tegument protein from human cytomegalovirus is the founding member of a new deubiquitinase family[24]. To this day, the scope of this family has been limited to a set of closely related proteins from vertebrate herpesviruses. By performing a comprehensive bioinformatical screen, followed by biochemical and structural characterization of interesting candidates, we were able to identify active members of this family in a wide range of eukaryotic species and a few host-associated bacteria. Due to the similarity to the herpesvirus proteins, we refer to this extended enzyme family as VTD (for <u>V</u>iral <u>T</u>egument-like <u>D</u>eubiquitinases). Our data suggest an origin of the VTD family by branching off from a non-DUB protease family, followed by a change in active site topology. Several pieces of evidence suggest that VTD proteins have spread to other taxa by becoming incorporated into DNA transposons and later 'domesticated' in multiple independent events. During this process, subfamilies of VTDs evolved to assume different linkage specificities, highlighting the versatility of this unusual deubiquitinase class.

## Results

### Bioinformatical discovery of VTD homologs

In order to learn more about the origins and evolution of the enigmatic herpesviral tegument DUBs, we used bioinformatical screens to identify distant homologs (Supplementary Fig. 1). In the first step, a multiple alignment of the catalytic domains of all recognizable herpesviral homologs of UL36 (Herpes Simplex Virus), UL48 (Human Cytomegalovirus) and BPLF1 (Epstein-Barr Virus) was generated by using MAFFT[31]. The resulting alignment was converted into a generalized sequence profile[32] and used to search current protein databases. After two cycles of iterative profile refinement, a set of 2951 significantly matching sequences was retrieved, among them 1730 sequences from herpesviruses and 1221 non-viral sequences. Sequence analysis of the latter group revealed that the majority of non-viral homologs are found within the open reading frame of Helitrons, a class of rolling-circle DNA transposons present in all eukaryotic kingdoms[33]. Autonomous Helitron elements lack the classical transposase/integrase found in other DNA transposons, but rather encode a so-called 'RepHel' protein with both HUH-endonuclease and 5' to 3' helicase activity[34]. The Helitron members hosting the recognizable VTD domains carry an additional C-terminal AP-type endonuclease and thus belong to the Helentron subclass of rolling-circle transposons[34]. Previously, Helitrons have been described to sometimes include OTU-like domains, although their potential deubiquitinase activity has never been addressed[34]. In fact, 121 of the VTD-containing Helitrons were found to also contain OTU domains. With the exception of several pseudogenized sequences, the tegument-like domains of presumed autonomous Helitrons have all catalytic residues conserved, suggesting that these transposon-associated domains have DUB activity. VTD-containing Helitrons were identified in vertebrates, insects, cnidarians, and molluscs. For further biochemical and structural characterization, we selected two Helitrons from zebrafish: DrT1 with two VTD domains but without OTU (LOC108180207 on chromosome 20) and DrT2 with a single active VTD and an additional OTU domain (AL772241 on chromosome 25) (Fig. 1a, b).

A second large group of non-viral VTD domains was identified in insects outside of a recognizable transposon context. Although these VTD domains are closely related to the Helitron-encoded VTDs (Supplementary Fig. 2), they are likely devoid of enzymatic activity as they lack one or more of the active site residues. Many of the identified insect sequences carry multiple copies of the inactive VTD (iVTD) domains, including three of the five identified proteins from *Drosophila melanogaster*: CG32436 (5 copies), CG939 (3 copies), CG4669 (2 copies), CG14402 (1 copy), CG32462 (1 copy). Interestingly, all five drosophila proteins are annotated to have a testis-specific expression pattern[35].

Besides the helitrons and inactive insect proteins, the list of identified VTD homologs also contains a few bacterial sequences, including Wc-VTD1 (wcw_1294) from *Waddlia chondrophila*, an

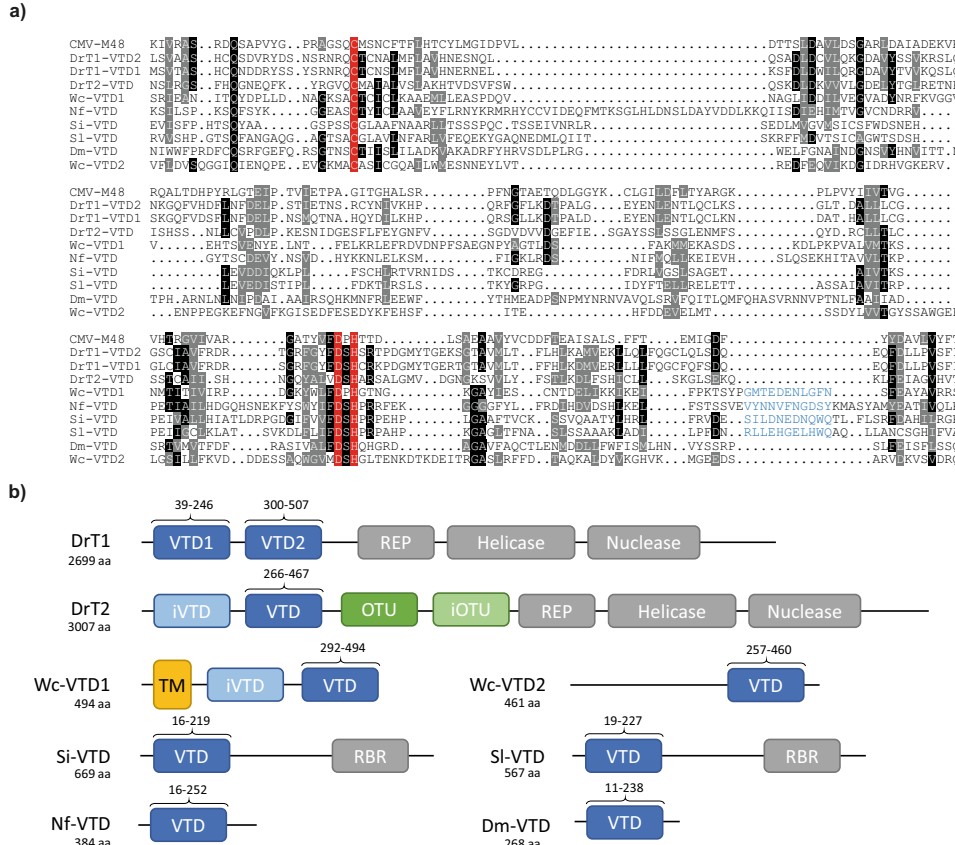

**Fig. 1 | The expanded VTD family. a** Structure-guided alignment of representative members. Residues printed on black or grey background are invariant or conservatively replaced in at least 50% of the sequences. The active site residues are highlighted in red, the loop associated with K6-specificity is shown in blue. **b** Domain architecture of the proteins analyzed in this work. Active and inactive VTD domains are shown in blue, other DUB-domains in green, transmembrane-regions in orange and other domain types in grey. The position of the VDT domains as aligned in part a) of the figure are indicated by numbers.

emerging pathogen linked to miscarriages and infertility[36,37]. While the bacterial sequences reach a significant similarity score, they are far more divergent from the viral VTDs (Fig. 1a, Supplementary Fig. 2). Their relationship to the established VTDs could be corroborated by Hidden-Markov-Model (HMM) searches starting from the bacterial members, which also uncovered a second, particularly divergent VTD in the *Waddlia chondrophila* gene Wc-VTD2 (wcw_1327) – a protein that did not reach significant scores in the initial search. Both *Waddlia* VTDs were selected for further biochemical analysis, their domain architecture is shown in Fig. 1b. The inclusion of the bacterial sequences into HMM searches revealed a further VTD subfamily with members in the fungal class *Agaricomycetes*, as well as in several protist classes. The fungal and protist VTD domains are rather similar to each other, but quite different from the viral and transposon-encoded VTDs (Supplementary Fig. 2). All members show active site conservation, and we selected two fungal members from *Serendipita indica* (PIIN_06337) and *Serpula lacrymans* (LOC18815922), and one protist member from the pathogenic amoeba *Naegleria fowleri* (FDP41_009013) for further analysis. Their domain structure is shown in Fig. 1b.

Finally, an HMM constructed from all VTDs found so far was run against protein family databases of various species, using the HHsearch algorithm[38]. In this search, the *Drosophila melanogaster* male sterile gene ms(3)76Ca (CG14101) and five uncharacterized genes from the model nematode *Caenorhabditis elegans* (C09H10.9, C18G1.9, C14C11.1, ZC317.6, Y53F4B.36) were identified, which are broadly conserved in other insects and nematodes and do not form part of transposons. The coding sequences of these genes are restricted to a single VTD domain (Fig. 1a, b), which

contains all predicted catalytic residues. As shown in Supplementary Fig. 2, this invertebrate subfamily is quite divergent and shows no particular relationship to the insect-encoded inactive VTDs. We selected the Drosophila ms(3)76 Ca sequence for further biochemical analysis[39].

Ubiquitin is highly conserved in all eukaryotic species harboring VTD-type deubiquitinases, with only two or three amino acid changes between human ubiquitin and its *Agaricomycete* and *Naegleria* homologs, respectively. Since the few points of divergence are distant typical DUB cleavage or recognition sites, we performed all deubiquitinase assays with human ubiquitin.

### Helitron-encoded VTDs have similar activities as herpesviral deubiquitinases

The three intact VTD domains encoded by the two selected *Danio rerio* helitrons were expressed in *E. coli*, purified, and tested against a set of model substrates comprising C-terminally propargylated ubiquitin and ubiquitin-like modifiers. These activity-based probes use a C-terminal PA-group to react covalently with functional thiol proteases of the correct specificity, resulting in an adduct of increased molecular weight[40]. The zebrafish VTDs DrT1-VTD1 and DrT1-VTD2, both derived from the same Chr.20 helitron copy, reacted readily with Ub-PA and NEDD8-PA, while DrT2-VTD from the Chr.25 helitron copy only reacted with the propargylated ubiquitin probe (Fig. 2a–c). No reaction with the other three tested probes SUMO1-PA, SUMO3-PA and ISG15[CTD]-PA (C-terminal domain of ISG15) could be detected. To directly compare the ubiquitin- and NEDD8-directed activities of the helitron-VTDs, they were additionally incubated with fluorogenic model substrates Ub-AMC and NEDD8-AMC, both carrying a C-terminal cleavable 7-amino-

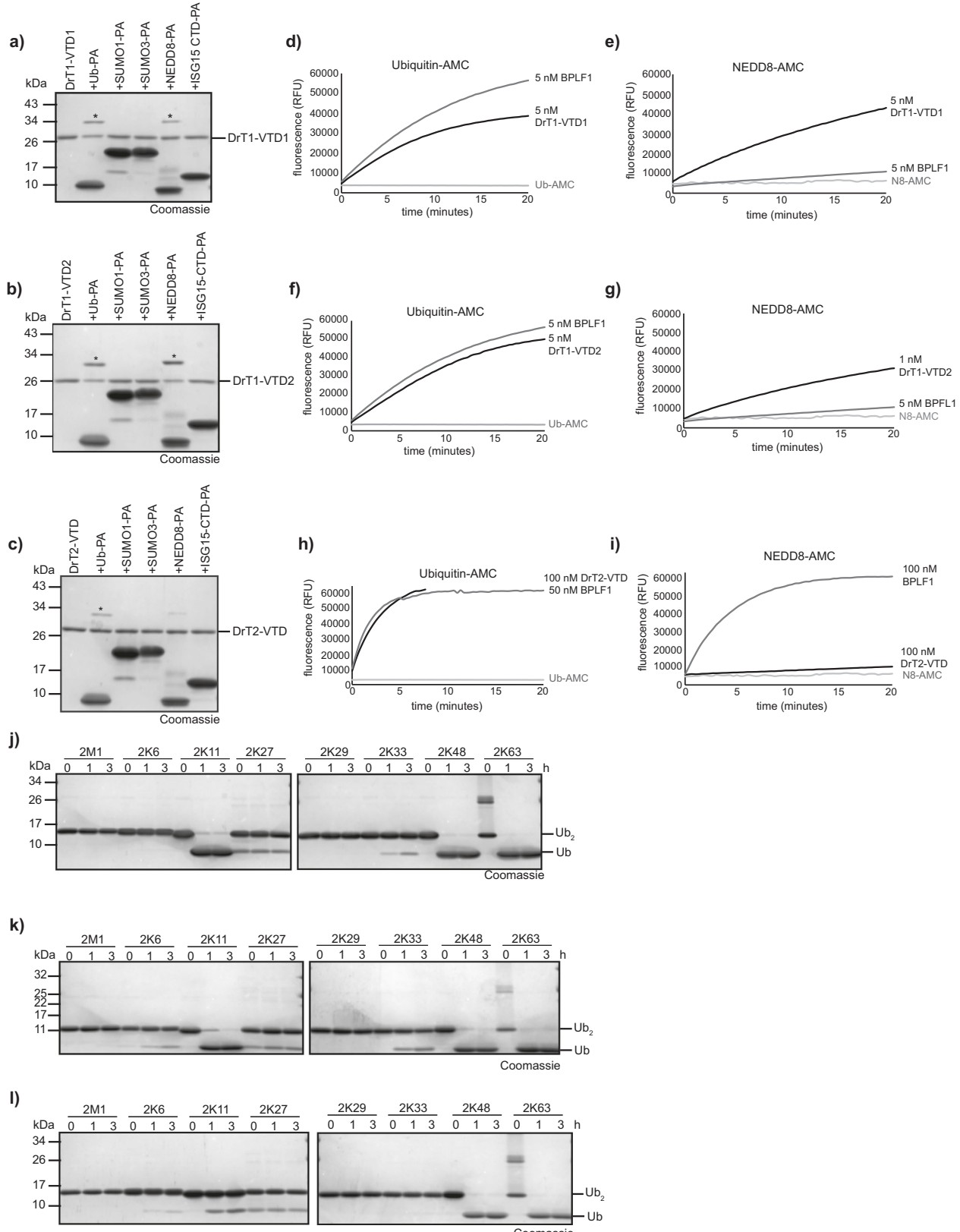

**Fig. 2 | Activity of Helitron-encoded VTDs. a–c** Reaction of DrT1-VTD1 (**a**), DrT1-VTD2 (**b**) or DrT2-VTD (**c**) with Ub and Ubl activity-based probes. Asterisks mark the shifted bands after reaction. **d, e** Activity of 5 nM DrT1-VTD1 or BPLF1 against Ubiquitin-AMC (**d**) and NEDD8-AMC (**e**). The RFU values are the means of tripli-cates. **f, g** Activity of 5 nM DrT1-VTD2 or BPLF1 against Ubiquitin-AMC (**f**) and 1 nM Dr1VTD2 against NEDD8-AMC (**g**). The RFU values are the means of triplicates.

**h, i** Activity of 100 nM DrT2-VTD or 50 nM BPLF1 against Ubiquitin-AMC (**h**) and of 100 nM DrT2-VTD or BPLF1 against NEDD8-AMC (**i**). The RFU values are the means of triplicates. **j–l** Linkage specificity analysis of DrT-VTDs. A panel of homotypic di-ubiquitin chains was incubated with DrT1-VTD1 (**j**), DrT1-VTD2 (**k**) or DrT2-VTD (**l**) for the indicated time points. Source data are provided as a Source Data file.

4-methylcoumarin (AMC) group. DrT1-VTD1 and DrT1-VTD2 were similarly efficient in removing ubiquitin and NEDD8 from the C-terminally fused fluorophore, as indicated by an increase in fluorescence. Compared to viral BPLF1, the DRT1-VTDs were similarly active towards Ub-AMC but cleaved Nedd8-AMC more efficiently than BPLF1 (Fig. 2d–g, Supplementary Fig. 3a). DrT2-VTD, derived from another helitron, was also active against the ubiquitin-AMC substrate, albeit with markedly lower reactivity compared to the other two helitron VTDs and BPLF1, while its ability to cleave NEDD8-AMC was negligible (Fig. 2h-i). To further investigate the DUB activity and linkage preferences of the three newly discovered deubiquitinases, the three enzymes were tested against a panel of di-ubiquitin species comprising all eight homotypic chain types. All three DrT-VTDs preferentially cleave K48- and K63-linked di-ubiquitin chains, with DrT1-VTD1 and DrT1-VTD2 also showing substantial activity towards K11-linked chains (Fig. 2j–l, Supplementary Fig. 3b–d). These results show that the three helitron-derived VTDs have a similar specificity as the viral tegument deubiquitinase UL36 from HSV-1, which has been shown to prefer K11-, K48- and K63-linked chains[41].

## A Helitron-VTD structure

To further elucidate the structural basis for VTD activity, we solved the crystal structure of DrT1-VTD2 (residues 300 to 520 of Uniprot entry A0A2R8QBC4) at a resolution of 1.90 Å (Table 1) and compared it to the ubiquitin complex of mouse cytomegalovirus M48, the only available structure of a herpesviral tegument deubiquitinase. As shown in Fig. 3a, the asymmetric unit contained one DrT1-VTD2 domain, which was completely resolved with the exception of residues 107 to 109 located in a protruding loop where Leu108 and Gly109 have rather weak density. In total, the structure contains four α-helices and nine β-strands, which adopt the form of an α-β-α sandwich, in which a bifurcated antiparallel β-sheet is flanked by α-helices. The helitron-VTD structure could be superimposed with the M48 catalytic domain (pdb:2J7Q) with an RMS distance of 2.5 Å over 192 residues, demonstrating an almost identical fold (Fig. 3b). The only major difference was observed for the aforementioned protruding loop without secondary structure elements of DrT1-VTD2, which in M48 corresponds to a well-folded β-hairpin contacting the outgoing S1-ubiquitin[24]. Since the DrT1-VTD2 structure does not contain ubiquitin, the β-hairpin structure might be induced by substrate binding rather than representing a real difference between viral and transposon-encoded VTD domains. While viral and transposon-encoded VTDs adopt the papain-like fold shared with all other thiol deubiquitinases, they have a highly unusual active site architecture (Fig. 3c). In VTDs, the catalytic histidine and aspartate residues form an invariable Asp-x-His motif, while in typical DUBs the catalytic Asp/Asn residue follows the His residue at a variable distance. As a consequence, the catalytic histidine (His453 in DrT1-VTD2 and His158 in M48) resides on the opposite site of the catalytic triad, creating a mirror image to the classic papain fold active site (Supplementary Fig. 3e–g). In the herpesviral M48 structure, a second histidine (His141) is positioned close to the catalytic site and was found to act as a partial replacement for the main catalytic residue His-158[24]. By contrast, the helitron VTD structure does not contain a second histidine residue with the potential to be involved in catalysis.

While in terms of sequence similarity and active site architecture, the VTDs are quite distinct from other deubiquitinases and even other cysteine proteases, a structure similarity search of the DrT1-VTD2 using the DALI method[42] revealed an interesting relationship to bacterial YopT-like enzymes. This family contains non-DUB proteases – often targeting small G-proteins – as demonstrated for *Pseudomonas syringae* AvrPphB and its homolog YopT from *Yersinia pestis*[43]. The structural alignment of DrT1-VTD2 with AvrPphB (pdb:1UKF) gave an RMS of 5.0 Å over 160 residues and shows an overall similar fold, while highlighting the different active

site architecture (Fig. 3d, e), since the YopT family uses the conventional Cys-His-Asp catalytic triad. To confirm the predicted role of Cys322, Asp451 and His453 as the catalytic triad of DrT1-VTD2, these residues were mutated to alanine and tested for their catalytic

**Table 1 | Data collection and refinement statistics**

| | DrT1-VTD2 | Wc-VTD1 | Wc-VTD1–Ubiquitin |
|---|---|---|---|
| Wavelength (Å) | 0.9747 | 0.9763 | 1.0000 |
| Resolution range (Å)[a] | 42.37 – 1.90 (1.97 – 1.90) | 51.06 – 1.70 (1.80 – 1.70) | 50.88 – 1.73 (1.84 – 1.73) |
| Space group | C 2 2 2$_a$ | P 1 2$_a$ 1 | P 3$_b$ 2 1 |
| Unit cell (Å, deg) | 119.01 120.67 34.17 90 90 90 | 37.12 70.30 75.89 90 101.84 90 | 101.77 101.77 63.20 90 90 120 |
| Total reflections | 481,803 (14,406)) | 279,246 (41,211) | 690,794 (107,527) |
| Unique reflections[b] | 37,458 (2724) | 81,659 (13,147) | 39,674 (6,344) |
| Multiplicity[b] | 12.9 (5.3) | 3.4 (3.2) | 17.4 (16.9) |
| Completeness (%) | 100.0 (99.9) | 98.8 (98.3) | 100.0 (100.0) |
| Mean I/sigma(I) | 11.5 (1.7) | 8.5 (1.0) | 11.0 (1.1) |
| Wilson B-factor | 23.3 | 25.9 | 30.6 |
| R-merge | 0.198 (1.02) | 0.079 (1.18) | 0.146 (2.57) |
| R-meas | 0.209 (1.09) | 0.106 (1.6) | 0.148 (2.78) |
| CC1/2 | 0.998 (0.63) | 0.995 (0.63) | 0.998 (0.50) |
| Refinement resolution range | 42.37 – 1.90 (1.97 – 1.90) | 51.06 – 1.70 (1.76 – 1.70) | 50.88 – 1.73 (1.79 – 1.73) |
| Reflections used in refinement | 19,941 (1930) | 41,742 (4124) | 39,633 (3914) |
| Reflections used for R-free | 1494 (144) | 2075 (208) | 1985 (196) |
| R-work | 0.167 (0.257) | 0.205 (0.382) | 0.167 (0.320) |
| R-free | 0.213 (0.283) | 0.244 (0.393) | 0.192 (0.373) |
| CC (work) | 0.96 (0.87) | 0.96 (0.80) | 0.968 (0.74) |
| CC (free) | 0.93 (0.89) | 0.96 (0.71) | 0.96 (0.68) |
| Number of nonhydrogen atoms | 1832 | 3382 | 2402 |
| macromolecules | 1674 | 3201 | 2246 |
| ligands | 0 | 0 | 28 |
| solvent | 158 | 181 | 139 |
| Protein residues | 212 | 406 | 284 |
| RMS (bonds) (Å) | 0.005 | 0.015 | 0.011 |
| RMS (angles) (degrees) | 0.85 | 1.18 | 1.22 |
| Ramachandran favored (%) | 96.14 | 99.24 | 98.92 |
| Ramachandran allowed (%) | 3.86 | 0.76 | 1.08 |
| Ramachandran outliers (%) | 0.00 | 0.00 | 0.00 |
| Rotamer outliers (%) | 0.00 | 0.57 | 0.40 |
| Clashscore | 3.62 | 6.17 | 4.66 |
| Average B-factor (Å[b]) | 31.06 | 43.90 | 39.46 |
| macromolecules | 30.42 | 43.92 | 38.91 |
| ligands | | | 66.66 |
| solvent | 37.87 | 43.41 | 45.07 |
| Number of TLS groups | 5 | 2 | 2 |
| PDB entry | 8ADD | 8ADC | 8ADB |

[a]Statistics for the highest-resolution shell are shown in parentheses.
[b]Friedel pairs are counted as different reflections for the anomalous data sets DrT1-VTD2 and Wc-VTD1.

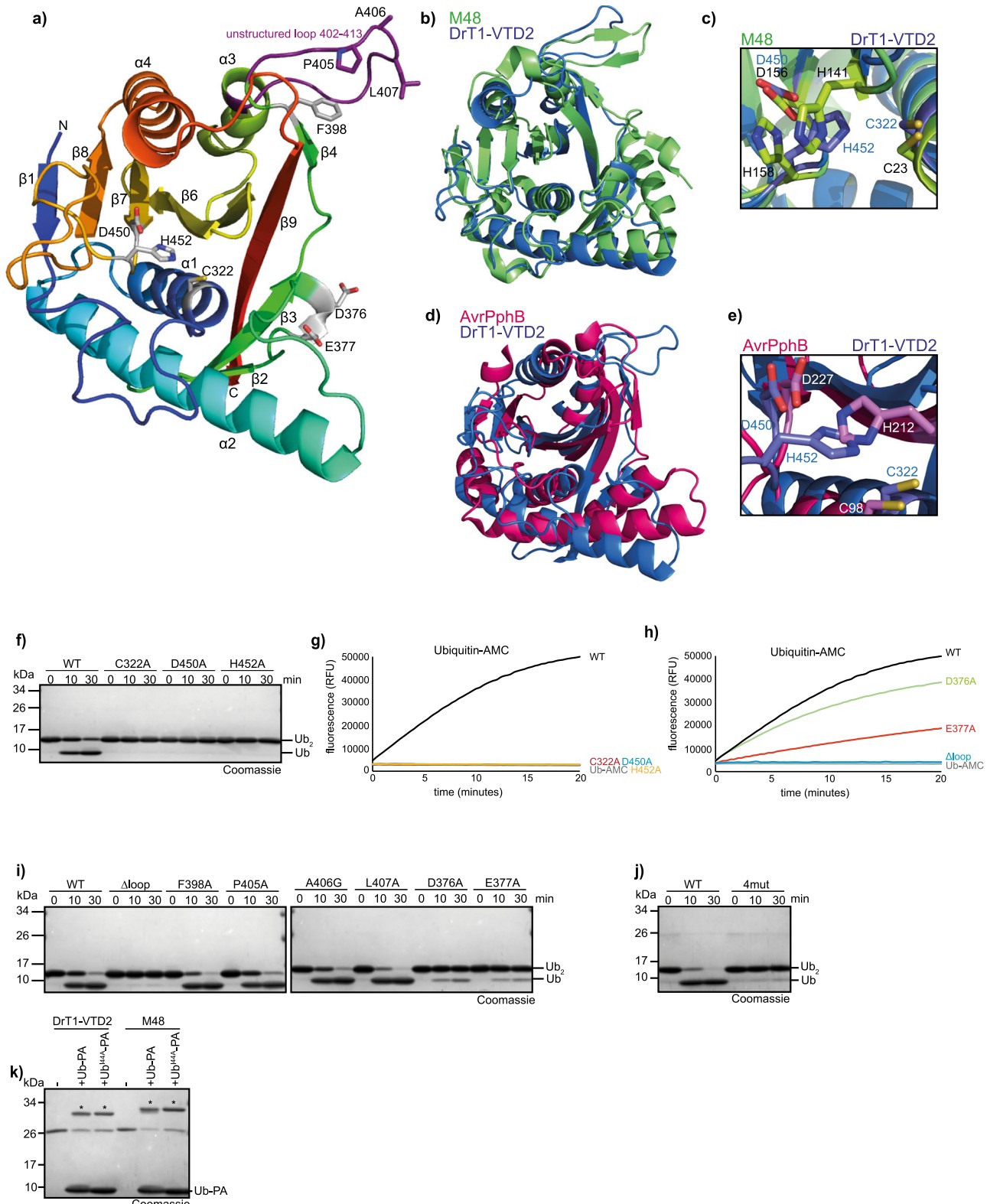

activity. Ubiquitin-AMC assays as well as incubation of K48-linked di-ubiquitin chains with C322A, D451A or H453A mutants demonstrated a complete loss of activity (Fig. 3f, g).

Since all attempts to crystallize DrT1-VTD2 in complex with ubiquitin failed, the structural correspondence to M48 was used to identify potential ubiquitin-binding interfaces by superposition of the M48-Ubiquitin complex and DrT1-VTD2 (Supplementary Fig. 3h). In M48, a protruding β-hairpin consisting of residues 108 to 115 was

found to be important for ubiquitin binding, with Leu110 and Tyr113 making important contacts with His68 and Val70 of ubiquitin's Ile44 patch[24]. This β-hairpin corresponds to the flexible 402–413 loop in DrT-VTD2. However, the β-hairpin region is not well conserved among VTD enzymes, making it difficult to identify corresponding residues. When the DrT1-VTD2 residues 403–411 were replaced by a short Gly-Gly-Ser linker, the enzymatic activity against Ub-AMC and di-ubiquitin was completely lost (Fig. 3h, i). By contrast, single point mutations of three

**Fig. 3 | Crystal structure of a Helitron-encoded VTD. a** Crystal structure of DrT1-VTD2 in cartoon representation. The catalytic triad and residues potentially important for ubiquitin interaction are shown as sticks. The unstructured loop is coloured purple. **b** Structural superposition of DrT1-VTD2 (blue) with M48 (PDB: 2J7Q, green). The RMSD is 2.5 Å for 192 $C_\alpha$ atoms. **c** Active site architecture of DrT1-VTD2 (green sticks) compared to M48 (PDB: 2J7Q, purple sticks). **d** Structural superposition of DrT1-VTD2 (blue) with AvrPphB (PDB: 1UKF, pink). The RMSD is 5.0 Å over 160 residues. **e** Active site architecture of DrT1-VTD2 (green sticks) compared to AvrPphB (PDB: 1UKF, purple sticks). **f** Activity of wildtype DrT1-VTD2 (WT) against K48-linked di-ubiquitin, compared to the catalytically inactive mutants C322A, D451A and H453A. **g** Activity of 5 nM wildtype DrT1-VTD2 (WT, black) against Ubiquitin-AMC, compared to 5 nM of the catalytically inactive

mutants C322A (red), D451A (blue) and H453A (yellow). The RFU values are the means of triplicates. **h** Activity of 5 nM wildtype DrT1-VTD2 (WT, black) against Ubiquitin-AMC compared to 5 nM of the binding mutants D376A (green), E377A (red) and Δhairpin (blue). The RFU values are the means of triplicates. **i** Activity of wildtype DrT1-VTD2 (WT) against K48-linked di-ubiquitin compared to the binding mutants Δhairpin, F398A, P405A, A406G, L407A, D376A and E377A. **j** Activity of wildtype DrT1-VTD2 (WT) or the quadruple mutant DrT1-VTD2[4mut] (F398A, P405A, A406G and L407A) against K48-linked di-ubiquitin. **k** Reaction of DrT1-VTD2 or M48 with the wildtype Ub (Ub-PA) and Ub[I44A] activity-based probe after overnight incubation (18 h). Asterisks mark the shifted bands after reaction. Source data are provided as a Source Data file.

potential contact residues within the loop (P405A, A406G and L407A) and one residue in the adjacent β4 strand (F398A) showed little effect on the chain-cleaving activity (Fig. 3i). However, the quadruple mutant DrT1-VTD2[4mut] (F398A, P405A, A406G and L407A) completely lost its activity against di-ubiquitin, suggesting a complex multi-factorial recognition of the ubiquitin Ile44 patch (Fig. 3j). To further assess the importance of direct Ile44 recognition, which is absent in the M48-Ub complex structure, DrT1-VTD2 and M48 were incubated with a ubiquitin activity-based probe, in which Ile44 was mutated to alanine (Ub[I44A]-PA). Both enzymes reacted with this mutant probe as well as with wildtype Ub-PA (Fig. 3k), suggesting a shared lack of direct Ile44 recognition by these DUBs.

As a second substrate recognition surface, typical DUBs bind the C-terminal R-x-R motif of the S1 ubiquitin by one or two salt bridges using acidic residues of the DUB. Based on their conservation pattern in the VTD family and their positioning within the structure, Asp376 and Glu377 might be suitable candidates for this interaction. Individual mutations of both residues (D376A and E377A) resulted in a strongly reduced chain cleaving activity (Fig. 3i), and also a reduced cleavage of Ub-AMC, the latter effect being more pronounced for E377A (Fig. 3h). This stabilization of ubiquitin's C-terminus enables Dr1VTD2 to cleave peptidic RLRGG-AMC, an activity that was either strongly reduced (D376A) or completely abrogated (E377A) in the respective mutant (Supplementary Fig. 3i).

## Bacterial VTD domains show linkage-specific ubiquitin cleavage

While the viral and helitron-encoded VTD domains resemble each other in sequence, structure and specificity, the bacterial VTDs, of which Wc-VTD1 is a typical example, are more divergent. The second Waddlia-encoded Wc-VTD2 is particularly divergent and forms an outlier even to the other bacterial VTDs (Supplementary Fig. 2) Under the ORF name wcw_1327, the Wc-VTD2 had previously been described as an immunogenic protein[44]. We analyzed the catalytic domain (aa279-494) of Wc-VTD1 and the full-length Wc-VTD2 for their ability to cleave model substrates and ubiquitin chains. Both Waddlia VTDs reacted with the activity-based Ub-PA probe, but not with the analogous UBL probes SUMO1-PA, SUMO3-PA, NEDD8-PA or ISG15CTD-PA (Fig. 4a, b). When tested for their catalytic activity against AMC substrates, neither Wc-VTD1 nor Wc-VTD2 were able to cleave Ub-AMC or NEDD8-AMC (Fig. 4c, d). In the di-ubiquitin cleavage assay, Wc-VTD1 showed a strong preference for K6-linked chains. Apart from some trace activity against K11-diUb, none of the other linkages showed any cleavage in this assay (Fig. 4e). By contrast, Wc-VTD2 preferentially hydrolyzed K63-linked chains, with minor activity towards K11-linked di-ubiquitin (Fig. 4f). The strong linkage specificity of the two Waddlia VTDs suggests an important contribution of the proximal (S1') ubiquitin binding site and might thus explain the lack of activity against the mono-Ub based AMC substrate (Fig. 4c). Summarizing, the bacterial VTDs do not only diverge in sequence conservation from their viral counterparts, they have also acquired a different specificity, possibly accompanied by a different ubiquitin-recognition mode.

## Structures of Wc-VTD1 reveal an unusual Ub binding mode

In order to gain insight into the functional differences of bacterial VTDs in comparison to their relatives from viruses and transposons, the structure of Wc-VTD1 was solved in the unbound form at 1.8 Å resolution as well as in a covalent complex with ubiquitin at 1.9 Å resolution. The latter was achieved by reacting the protease with ubiquitin-PA, which traps the Wc-VTD1 in an intermediate state with the ubiquitin being bound to the S1 site in a covalent linkage to the active cysteine. The tertiary structure contains nine α-helices and seven β-strands, adopting a fold with overall similarity to the viral and transposon-encoded VTDs (Fig. 5a, Supplementary Fig. 4a). The catalytic triad is formed by Cys319, Asp440, and His442. As previously observed in the M48 crystal structure[24], the conformation of the catalytic histidine appears as non-productive for catalysis due to the large distance to the cysteine (Supplementary Fig. 4b, c). However, considering the sequence alignment (Fig. 1a) and the site-directed mutagenesis experiments (Fig. 5h), the assignment of these residues to the catalytic triad is safe. The structures in the unbound and complexed state can be superimposed with an RMS distance of 0.538 Å over 144 Cα-atoms excluding some 40 residues, thus indicating little overall conformational changes upon binding to the Ub-PA substrate. However, some parts of Wc-VDT1 undergo relatively large movements upon substrate binding. Helix α9 (residues 474 to 486) and its preceding loop (residues 469 to 476) are pushed about 9 Å toward the active site by ubiquitin's loop consisting of residues 7 to 11, where it offers an interaction surface for the Ile-36 patch of the S1 ubiquitin (Fig. 5a–c). On the opposite side, helix α1 (residues 302 to 305) together with the following amino acids 306 to 311 is also shifted by a distance of about 8 Å. However, helix α1 itself does not seem to be involved in direct interactions with the bound substrate. It is located on the surface rather distant from the active site and is involved in the crystal contacts between the two crystallographically independent molecules in the apo structure. Roughly at this position, the proximal (S1') ubiquitin would be placed and thus helix α1 may be important for linkage specificity. The β-hairpin turn consisting of residues 421 to 423 is pushed away by about 2 Å from the active site cleft by ubiquitin's four very C-terminal residues (Fig. 5b). To compare the ubiquitin interaction of Wc-VTD1 to viral VTDs, a superposition of the bacterial protease with M48 was created, yielding an RMSD of 1.76 Å over 1113 atoms (Supplementary Fig. 4d). Wc-VTD1 does not contain the ubiquitin-interacting β-hairpin of its viral relatives, but rather an irregular loop comprising residues 380–386 (colored yellow Fig. 5a, d and Supplementary Fig. 4d). This loop makes extensive contacts mainly via Pro383 and Phe384 to Leu8, Ile44, His68 and Val70 of the bound ubiquitin. The entire Ile44 patch of ubiquitin is localized within a hydrophobic pocket of the enzyme, which contributes multiple interactions between modifier and protease (Fig. 5d, Supplementary Fig. 4d).

Three residues were chosen to investigate their importance for ubiquitin recognition, based on their structural position: Phe384 located on the P383/F384 loop and Ala392 on the adjacent β2 strand both contact the Ile44 patch, while Leu481 is part of the mobile α9

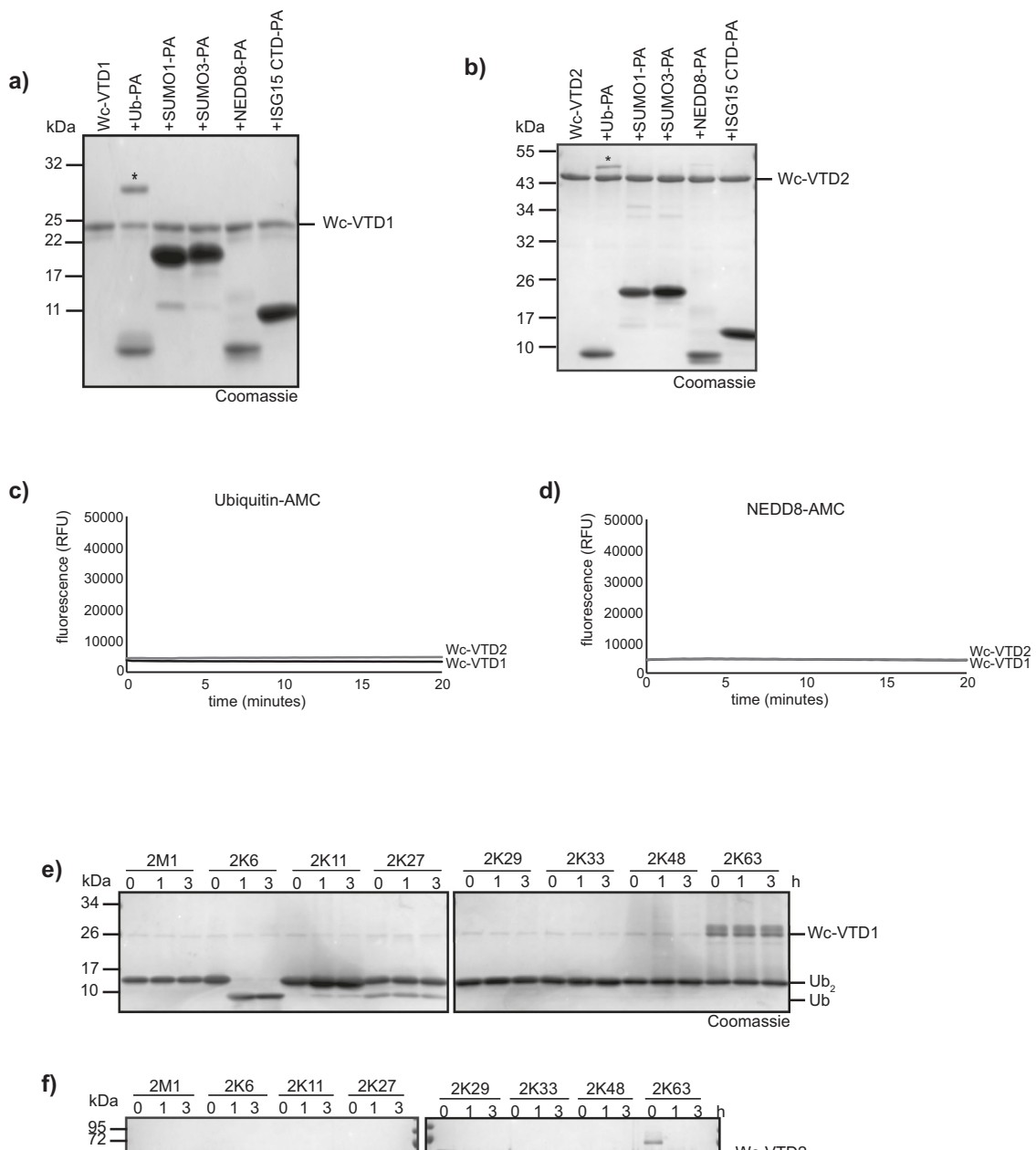

**Fig. 4 | Activity of bacterial VTDs. a**, **b** Reaction of Wc-VTD1 (**a**) and Wc-VTD2 (**b**) with Ub and Ubl activity-based probes. Asterisks mark the shifted bands after reaction. **c**, **d** Activity of Wc-VTD1 and Wc-VTD2 against Ubiquitin-AMC (**c**) and NEDD8-AMC (**d**). The RFU values are the means of triplicates. **e**, **f** Linkage specificity analysis of Wc-VTDs. A panel of homotypic di-ubiquitin chains was incubated with Wc-VTD1 (**e**) or Wc-VTD2 (**f**) for the indicated time points. Source data are provided as a Source Data file.

helix and contacts the Ile36 patch of ubiquitin. All three tested mutations (F384A, A392G and L481A) strongly reduce the activity of Wc-VTD1 on K6-linked ubiquitin chains (Fig. 5e). While the F384A and A392G mutants showed some residual activity, cleavage was completely abrogated by the L481A mutation, demonstrating the importance of the Ile36-patch recognition. To test if Wc-VTD1 shares with the helitron-VTD the property that the Ile44 patch of ubiquitin is contacted, while Ile44 itself is not crucial, the Wc-VTD1 domain was also incubated with the Ile44-mutated activity-based probe Ub$^{I44A}$-PA. Unlike the transposon-encoded enzyme, Wc-VTD1 only reacted with the wildtype Ub-PA, showing that this enzyme depends on a recognition of Ile44 itself (Fig. 5f). The alignment of Wc-VTD1 with M48 and other VTDs (Fig. 1a) suggested Glu363 and Asn364 as potential salt bridge partners for the basic RxR motif at the ubiquitin C-terminus. Mutation of either residue indeed had a strong impact on Wc-VTD1 activity against K6-linked chains: The E363A mutant appeared to be as

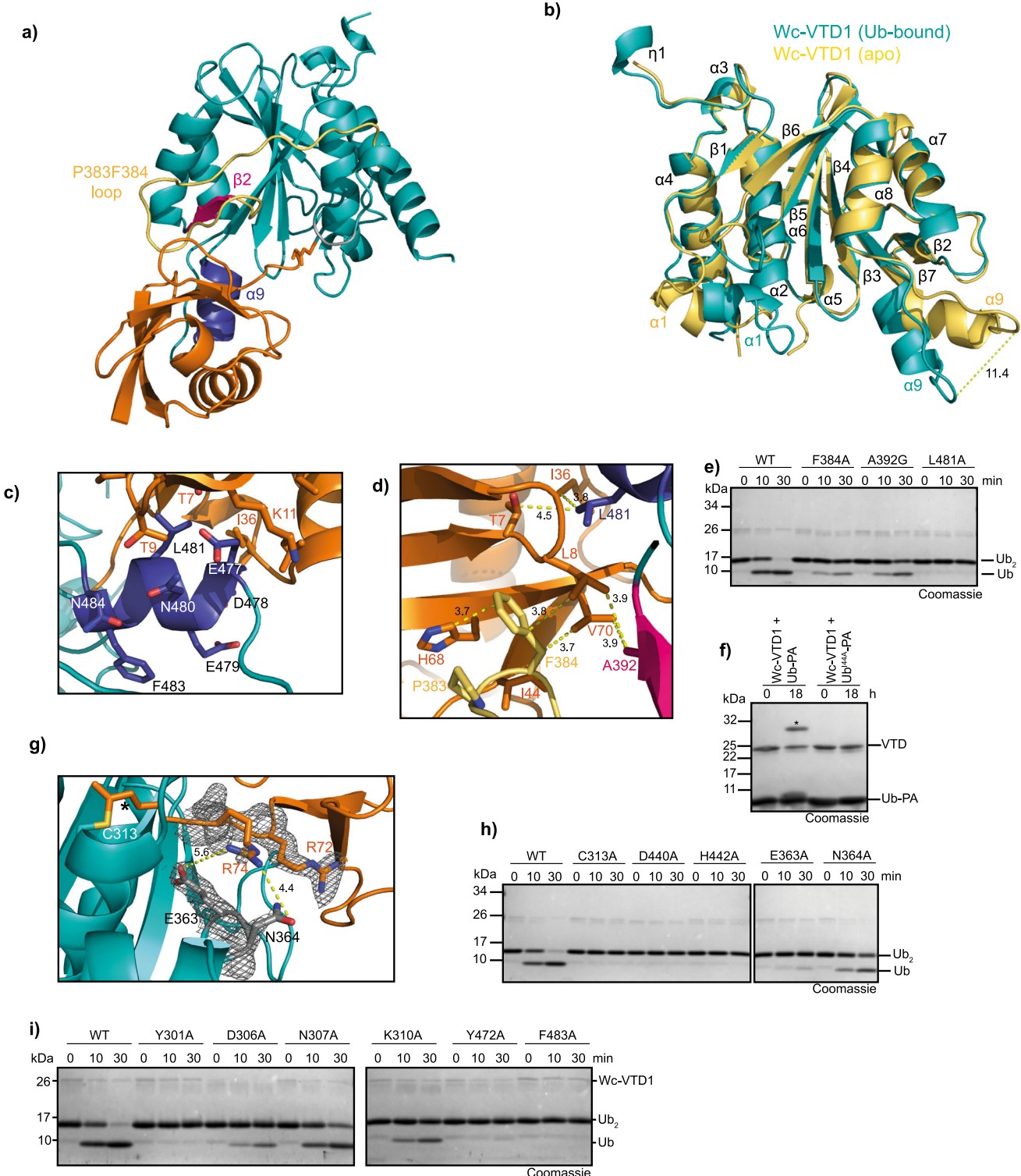

**Fig. 5 | Crystal structure of *Waddlia* VTD1. a** Crystal structure of Wc-VTD1 (teal) in covalent complex with Ub-PA (orange) in cartoon representation. The α9 helix is colored blue, β2 is pink and the P383F384 loop is depicted in yellow. **b** Structural superposition of apo WcVTD1 (yellow) and ubiquitin-bound Wc-VTD1 (teal). The RMSD is 0.54 Å over 205 residues. **c** Magnification of α9 interaction with S1 ubiquitin. Residues of α9 in Wc-VTD1 are shown as blue sticks and the hydrophobic interaction interface of ubiquitin as orange sticks. **d** Putative interactions between ubiquitin's Ile44 patch (residues shown as orange sticks) with Wc-VTD1 through its α9 (blue), β2 (pink) and P383F384 loop (yellow). Mutated residues are shown as sticks. **e** Activity of wildtype Wc-VTD1 (WT) against K6-linked di-ubiquitin compared to the binding mutants F384A, A392G, L481A. **f** Reaction of Wc-VTD1 with the wildtype Ub (Ub-PA) and Ub[I44A] activity-based probe after overnight incubation (18 h). Asterisk marks the shifted bands after reaction. **g** Interaction of Wc-VTD1 (teal) with ubiquitin's C-terminus (orange). The C-terminus of ubiquitin is shown as orange sticks and the propargylamide bond is marked with an asterisk. The catalytic cysteine of Wc-VTD1 is colored yellow and E363 and N364 are shown as grey sticks. The electron density map highlights the flexible R74. **h** Activity of wildtype Wc-VTD1 (WT) against K6-linked di-ubiquitin compared to the catalytically inactive mutants C313A, D440A and H442A, and against binding mutants E363A and N364A. **i** Activity of wildtype Wc-VTD1 (WT) against K6-linked di-ubiquitin compared to mutants of surface-exposed residues, potentially forming the S1´ site. Source data are provided as a Source Data file.

inactive as the active site mutants, while the N364A mutation severely reduced K6-linked di-ubiquitin cleavage. (Fig. 5h). However, in the experimental crystal structure the side-chains of Asn364 of Wc-VTD1 and Arg74 of ubiquitin are completely disordered and there is no evidence for a significant interaction visible in the electron density map; the same is true for the moderately well-defined Arg72 and Glu363 side chains (Fig. 5g). A possible explanation for this is the very high ionic strength of the crystallization buffer (2 M citrate), which may interfere with the electrostatic interactions like salt bridges[45].

Due to on the absence of a proximal ubiquitin in the Wc-VTD1-Ub-PA complex structure, the mechanism of the K6-specificty remains elusive. Recently it was shown, that the K6-specificity of LotA is achieved by a structural rearrangement of the active site caused by Phe4 of the proximal ubiquitin[46]. However, Wc-VTD1 is able to cleave K6-linked di-ubiquitin harboring a F4A mutant in both moieties, indicating that this mechanism is not used (Supplementary Fig. 4e). In order to determine the S1´site of Wc-VTD1, several surface-exposed candidate residues were mutated and tested for cleavage of K6-linked di-ubiquitin (Supplementary Fig. 4f). Since the helices α1 and α9 both undergo conformational changes upon ubiquitin binding, they were of particular interest (Fig. 5b). Interestingly, single point mutations of Tyr301, Asp306 and Lys310, belonging to α1 and neighboring loops, strongly reduced the cleavage of di-ubiquitin. By contrast, mutation of Asn307, which is also part of α1 but faces away from the active site, did not alter the activity of Wc-VTD1 (Fig. 5i). The α9 helix was shown to be part of the S1 site, but the opposite, distal-ubiquitin averted face of this helix could potentially bind the proximal ubiquitin. Strikingly, mutations of Phe483 (α9) and Tyr472 (connecting loop) led to a complete loss of di-ubiquitin cleavage (Fig. 5i).

### Fungal and Naegleria VTDs are K6-directed deubiquitinases

Besides the transposon-derived and bacterial VTDs, potential candidates were also discovered in agaricomycete fungi and various protists. As fungal representatives, the VTD domain of *Serendipita indica* (Si-VTD) and *Serpula lacrimans* (Sl-VTD) were analyzed, while the short protein Nf-VTD from the pathogenic amoeba *Naegleria fowleri* was analyzed as protist representative. Since bacterially expressed Si-VTD and Sl-VTD become unstable upon tag removal, both proteins were purified with an N-terminal 6His-Smt3 tag. When incubated with a panel of UBL-derived activity-based probes, none of the three VTD candidates showed any reactivity (Fig. 6a–c). Similarly, activity against the fluorogenic substrates Ub-AMC and NEDD8-AMC was mostly absent, except for a weak cleavage of Ub-AMC by Nf-VTD (Fig. 6d–i). By contrast to their poor activity against mono-Ub-based substrates, all three tested VTD enzymes were able to cleave di-ubiquitin chains, with a marked preference for K6-linkages (Fig. 6j–l). Nf-VTD at 500 nM cleaved about half of the K6 chains within 3 hours, with minor activity against K48 chains. (Fig. 6j). Si-VTD required a higher concentration of 5 µM for reaching a similar level of cleavage (Fig. 6k), while Sl-VTD was more active at 500 nM and cleaved most of the K6-diUb after 1 h (Fig. 6l). Unlike the Naegleria enzyme, the two fungal VTDs were truly K6-specific and showed no reactivity against any other linkage type. Thus, in accordance with their sequence similarity to bacterial VTDs, the fungal and protist homologs share the K6-specificity observed for Waddlia Wc-VTD1 (Fig. 4e).

### Drosophila male-sterile VTD is a K48-directed deubiquitinase

As a representative for the divergent invertebrate-type VTDs found in insects and vertebrates, the protein encoded by the *Drosophila* Male sterile (3) 76Ca gene (Dm-VTD) was tested. As shown in Fig. 7a, the enzyme reacts with the Ub-PA probe, but also with a probe based on the C-terminal moiety of ISG15 – the latter being without biological relevance, since ISG15 is a vertebrate-specific modifier. No reactivity was observed towards NEDD8-PA or SUMO-PA probes (Fig. 7a). Dm-VTD was also shown to cleave the fluorogenic substrates Ub-AMC, but

not NEDD8-AMC (Fig. 7b, c). When testing Dm-VTD against a panel of di-ubiquitin substrates, a preferential cleavage of K48-linked chains was observed, with some reactivity towards K63 and K11 chains. Despite the large evolutionary distance (Supplementary Fig. 2), the enzymatic properties of Dm-VTD resemble those of viral and helitron-encoded VTDs.

## Discussion

VTD-type deubiquitinases are found in the genomes of animals, fungi, protists and bacteria, but their phyletic distribution is rather patchy. Horizontal gene transfer and recurrent gene losses are known to cause unusual phyletic distributions with different degrees of sparseness[47]. In the evolution of the VTD family, both mechanisms appear to be at play. There are several discontinuous taxa with broadly observed VTD genes: One of them are the *Herpesviridae*, which evolved 180–220 million years ago (mya)[48]. VTDs are found in α-, β-, and γ-herpesviruses, which infect mammals and birds, while no VTDs are found in *Alloherpesviridae*, a sister group that comprises fish and amphibian herpesviruses. Interestingly, A*lloherpesviridae* contain a large tegument protein with an OTU domain[49], suggesting that these viruses too might benefit from a ubiquitin-cleaving activity attached to the capsid. A second VTD-containing taxon is formed by the *Agaricomycetes*, a mushroom-forming class of *Basidiomycete* fungi. Most sequenced *Agaricomycete* genomes encode a single protein with a VTD-domain fused to an RBR-type ubiquitin ligase; in other genomes (e.g. the eponymous *Agaricus bisporus*) this gene is unannotated, but still present. The VTD-containing *Agaricomycetes* evolved approx. 185 mya, while the *Dacrymycetes*, which diverged ~40 million years earlier[50,51], encode the corresponding RBR gene without a VTD domain. Apparently, the acquisition of viral and fungal VTDs happened roughly simultaneously in the mesozoic era, around the advent of the first mammals.

Older VTD genes are found in the invertebrates. Almost all nematode genomes of clade III (*Ascaridomorpha*, *Spirumorpha*) and clade V (*Rhabditids*, *Strongylids*) contain VTD genes, while other nematode clades lack VTDs. A recent phylogenetic analysis joins clade III and V nematodes in one taxon, suggesting that a VTD gene was acquired by a common precursor of these clades – possibly 400–800 mya[52]. VTD-containing genes are also found in many arthropod clades, including insects (several flies, wasps, aphids, beetles), *Collembola* (several springtails) *Crustacea* (several ostracods), and *Myriapoda* (*Strigamia*). However, only very few species within these highly populated taxa do contain VTD genes, raising the question if this sparsity is the result of sweeping gene losses in most lineages, or if it is caused by multiple independent acquisitions – possibly from an arthropod-associated transposon source. The former idea is supported by dendrogram analysis of extant arthropod VTDs (Supplementary Fig. 5), showing that their similarity roughly follows the phylogenetic tree of the host organism. Other invertebrate VTD genes might have originated from transposons: Several *Cnidaria* (corals, sea anemones) contain multiple genes with VTD domains fused to various other protein regions – among them several transposon-associated domains. While VTD proteins are widespread in eukaryotes, bacterial VTDs are currently limited to two members of *Chlamydiales* and one *Myxobacterium*, arguing against a bacterial origin of this family.

Judging by the similarity of VTD structures to members of the YopT/HopN family, while also considering the unusual VTD active site architecture and lack of detectable sequence relationship to other DUB classes, it is likely that the first VTD proteases branched off a YopT-like precursor gene in some eukaryotic species. Since all characterized VTD proteins cleave ubiquitin, while known YopT/HopN proteases have other targets[43], this specificity shift might be a consequence of the 'flipped' active site topology. Interestingly, a similar change of active site topology with associated specificity shift has recently been described for members of the ZUFSP deubiquitinase

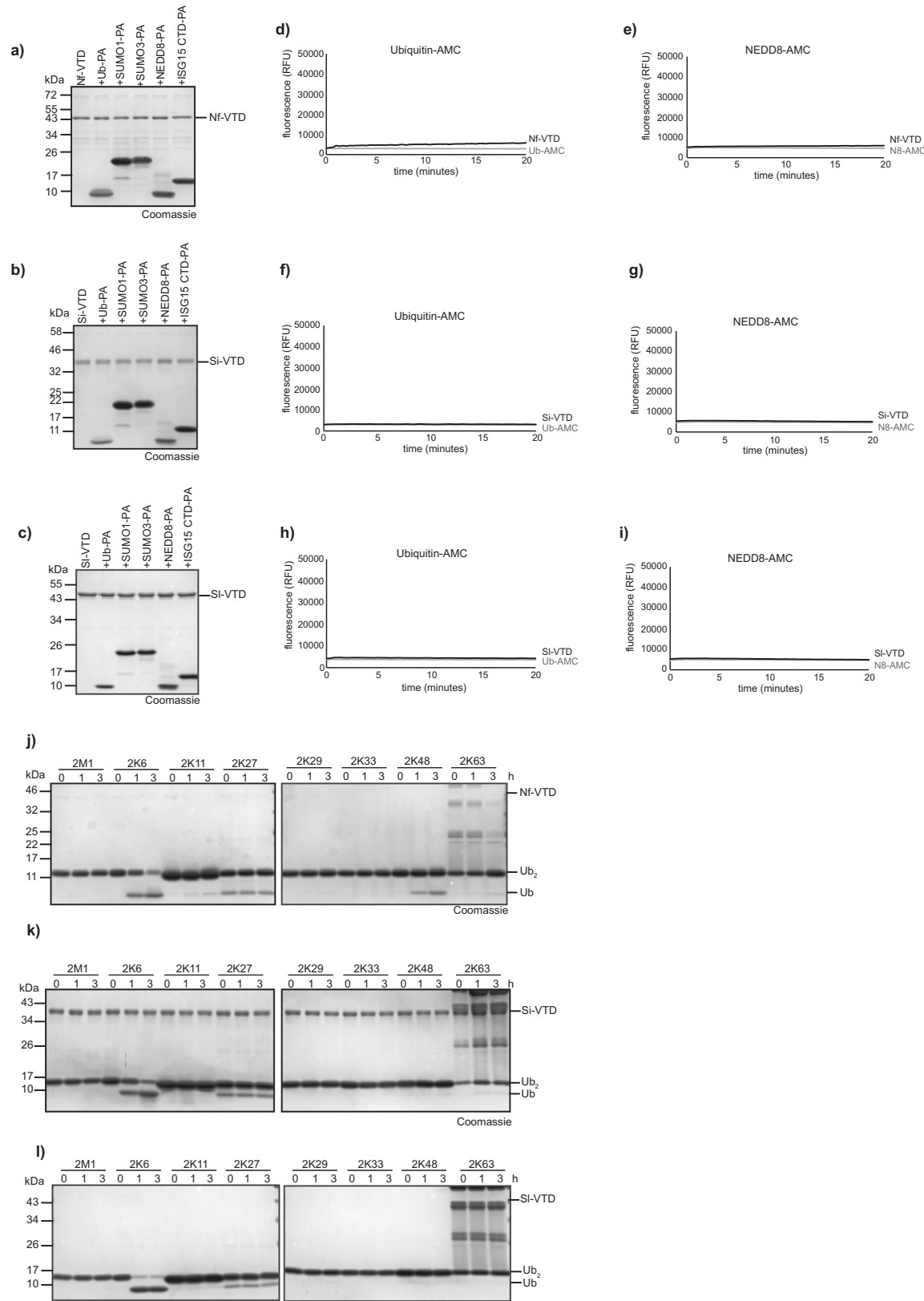

**Fig. 6 | Activity of fungal and protozoan VTDs. a–c** Reaction of Nf-VTD (**a**), Si-VTD (**b**) or Sl-VTD (**c**) with Ub and Ubl activity-based probes. **d**, **e** Activity of Nf-VTD against Ubiquitin-AMC (**d**) and NEDD8-AMC (**e**). The RFU values are the means of triplicates. **f**, **g** Activity of Si-VTD against Ubiquitin-AMC (**f**) and NEDD8-AMC (**g**). The RFU values are the means of triplicates. **h**, **i** Activity of Sl-VTD against Ubiquitin-AMC (**d**) and NEDD8-AMC (**e**). The RFU values are the means of triplicates. **j–l** Linkage specificity analysis of eukaryotic VTDs. A panel of homotypic di-ubiquitin chains was incubated with Nf-VTD (**j**), Si-VTD (**k**) or Sl-VTD (**l**) for the indicated time points. Source data are provided as a Source Data file.

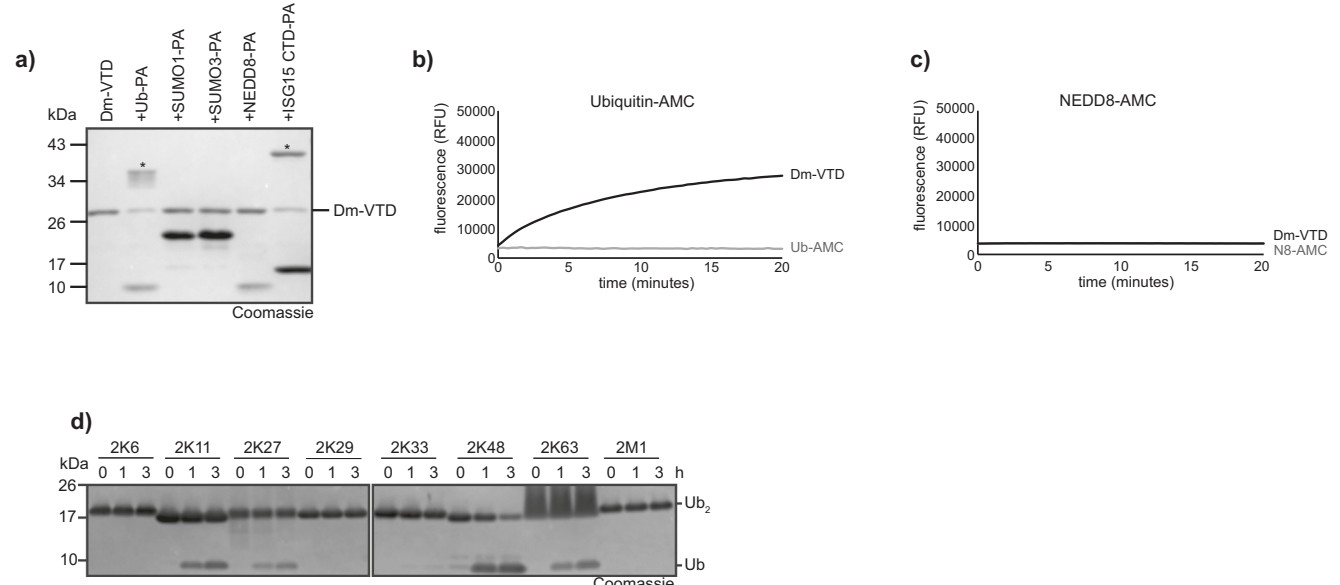

Fig. 7 | **Activity of invertebrate VTDs. a** Reaction of Dm-VTD with Ub and Ubl activity-based probes. Asterisks mark the shifted bands after reaction. **b-c** Activity of Dm-VTD against Ubiquitin-AMC (**b**) and NEDD8-AMC (**c**). The RFU values are the means of triplicates. **d** Linkage specificity analysis. A panel of homotypic di-ubiquitin chains was incubated with Dm-VTD for the indicated time points. Source data are provided as a Source Data file.

family: ZUP1 and other members of the ZUFSP family use the canonical active site architecture found in papain and YopT/HopN proteases, while the related UBL-specific proteases UFSP2 and ATG4 use a VTD-like catalytic triad including the D-x-H motif[53–56]. Apart from VTDs, members of the ZUFSP family are the only known deubiquitinases lacking the aromatic gatekeeper motif[19], suggesting a mechanistic similarity between these enzyme families.

There is insufficient data to fully elucidate the evolutionary history of the extant VTD family, but the following hypothesis (Supplementary Fig. 6) is plausible and compatible with the experimental data: After the 'birth' of the VTD family from a non-DUB source, it appears to have been co-opted relatively early by one or more transposon classes. Some of these transposons spread among invertebrates, leading to the introduction of VTDs into clade III/V nematodes and an early arthropod precursor. Judging by the extant Drosophila VTD, these early enzymes probably preferred K48 chains, suggesting a role in protecting proteins from proteasomal degradation. Later, the K6-specific VTD form evolved – possibly in some protist or associated transposon – and was subsequently introduced into the precursor of *Agaricomycete*s and several protist lineages, including *Excavata, and the precursor of Alveolata and Stramenopiles*. Protists are the most likely source for the bacterial K6-specific VTDs, since both *Waddlia* and *Simkania* are found in free-living amoebae[36,57]. At approximately the same time, the precursor of *Herpesviridae* co-opted a K48-selective VTD – probably from a Helitron-type rolling circle transposon, which became incorporated into the large tegument protein of vertebrate herpesviruses.

Due to our failure to co-crystallize VTD domains complexed with di-ubiquitin chains, we cannot pinpoint the structural basis for the different linkage-specificities of the K6-preferring VTDs from fungi, protists and bacteria and the K48/K63/K11-preferring VTDs from viruses and transposons. The latter group does not only efficiently cleave three different linkage types, but is also highly active towards Ub-AMC, suggesting that a binding of the proximal (S1') ubiquitin is not important for catalysis.

By contrast, the K6-specific VTDs are nearly inactive against Ub-AMC and might therefore harbor a specific recognition surface for the proximal (S1') ubiquitin in K6-orientation. The α1-helix and flanking regions of Wc-VTD1 are well-positioned to fulfill such a role. Since the α1-region is shifted markedly between free and ubiquitin-bound structures of Wc-VTD1 (Fig. 5b), a crosstalk between S1- and S1'-recognition appears possible. Indeed, the mutagenesis of several residues within the α1-region strongly impaired K6-cleavage, with Y310A being particularly effective (Fig. 5i). Since these residues line a partially hydrophobic surface next to the active site, a participation of Y301, D306, N307 and K310 in the recognition of the S1'-ubiquitin is likely. Beyond the S1'-ubiquitin recognition, the K6-specific Wc-VTD1 also recognizes the outgoing S1-ubiquitin via multiple surfaces: The poorly structured P383/F384 loop contacts the Ile44-patch of ubiquitin and appears to functionally replace the hairpin loop of viral and helitron-encoded VTDs. In addition, the Ile36 patch of ubiquitin is contacted by the C-terminal α9-helix of Wc-VTD1 (Fig. 5a), which is absent from the viral and helitron VTD structures. As highlighted in Fig. 1a, this helix shows little sequence conservation but is present in all VTDs with K6-specificity (Wc-VTD1, Nf-VTD, Si-VTD, Sl-VTD) while being absent from VTDs with other specificities. In a recent study on the K6-specific Legionella DUB LotA, an interaction between the Ile36 and Ile44 patches of two adjacent K6-linked ubiquitin moieties was reported, which had to be disrupted by the enzyme before cleaving the isopeptide bond[46]. The α9-helix of Wc-VTD1 appears to fulfills a similar task. Interestingly, mutagenesis of Y472 and F483, two α9 residues pointing away from the S1-ubiquitin, completely abrogated K6-chain cleavage (Fig. 5i). Since in the S1-occupied structure the α9 helix is shifted towards the α1 region (Fig. 5b), these residues (Y472 and F483) might join the α1 residues in S1'-recognition. In fact, the available structure and mutagenesis data suggest an attractive three-step model for explaining the K6 specificity of Wc-VTD1: In the apo-form, the α1 and α9 helices are far apart and the catalytic histidine residue is in a non-productive conformation. Upon binding of the S1-ubiquitin, the α1 and α9 helices are shifted towards each other, leading to the formation of a S1'-binding surface with contributions from both α1 and α9. However, the catalytic histidine is still not aligned properly with the active site; it is exposed on the surface, near the putative S1'-binding interface. Only upon binding of a S1'-ubiquitin with K6-linkage, the catalytic histidine will be pushed towards the active site and rendered productive.

Another open question concerns the functional role of VTD enzymes in their host species – and why many species groups apparently don't need these activities. Given that ubiquitination by K48-, K63- and K6-linked chains usually leads to very different outcomes, it is unlikely that all VTD deubiquitinases fulfil a common task. For the K48-cleaving herpesviral tegument DUBs, a number of biological effects have been published, often targeting the ubiquitin-mediated degradation of signaling components[23,25–27] or using the deneddylase activity to target cullin-based ubiquitin ligases of the host[28]. The closely related helitron-encoded VTDs might have a similar function by preventing the proteasomal degradation of proteins required for transposition. Interestingly, *Alloherpesviridae* – a virus group closely related to *Herpesviridae* – have replaced the VTD-domain in their tegument proteins by OTU-domains[49]. Similarly, several helitron-type rolling circle transposons also contain OTU domains besides – or instead of – VTD domains. Although neither viral nor transposon OTU domains have been tested for catalytic activity, those observations suggest that herpesviruses and helitrons benefit from a type of deubiquitination activity, which can be encoded by different DUB classes. Another intriguing observation is the male-specific expression of different animal VTD classes: There are five drosophila genes that encode proteins with inactive VTD domains closely related to the helitron-VTDs, all of which are expressed exclusively in the male testis[35]. The same is true for the *Drosophila* male sterile gene ms(3)76Ca (CG14101)[39], which encodes a very distantly related but enzymatically active VTD domain. This gene, like its homologs from the nematode *C. elegans*, is also expressed exclusively in the male germline[58]. Although the male-specific VTD types are not closely related, they both appear to be derived from domesticated transposons and either cleave K48-chains or are inactivated versions of K48-cleaving enzymes. It is therefore tempting to speculate that their transposon-encoded ancestors might have performed K48-linked deubiquitination in the male germline – possibly helping to overcome a mechanism for limiting transposon inheritance.

For the K6-specific VTDs, their biological role is equally enigmatic. In eukaryotes, no truly K6-specific DUBs are known; OTUD3 prefers K6 chains but also cleaves K11 linkages[13] and USP30 too cleaves K6-linkages[59] better than other chains. K6-linked ubiquitin chains have been implicated in various biological processes including DNA damage and mitophagy[7–9], but the particular role of this linkage type is insufficiently understood[6]. The only known example of a truly K6-specific deubiquitinase is the bacterial effector LotA from *Legionella pneumophila*, whose N-terminal OTU domain harbors this activity[15,46]. *Waddlia chrondophila* is unrelated to *Legionella*, but has a similar intracellular lifestyle within bacteria-containing vacuoles. Thus, an analogous role for the K6-specific LotA and Wc-VDT1 appears plausible – possibly through the evasion of ubiquitin-dependent autophagy after modification by the K6-specific ubiquitin ligase LRSAM1[10]. Besides the aforementioned herpesviral and helitron-related domain displacements, the LotA/Wc-VTD1 dichotomy might be a third case of co-opting non-homologous DUB types for fulfilling analogous tasks. It thus appears that VTD-type deubiquitinases are widespread and versatile enzymes, which can functionally replace OTUs – a DUB class known for their linkage-selective chain cleavage.

## Methods

### Sequence analysis

Sequence alignments were generated using the L-INS-I algorithm of the MAFFT package[31]. The multiple alignments were used for the generation of generalized profiles using pftools[32], and Hidden-Markov-Models using the HHSEARCH[38]. Generalized profile searches were performed iteratively, using all proteins from the Uniprot database (https://www.uniprot.org). Only database hits reaching a corrected p-value better than 0.01 were included into the next iteration cycle. Protein clustering was performed using the CLANS software[60].

Structure comparisons were performed using the DALI method[42] provided by the server under the URL http://ekhidna2.biocenter. helsinki.fi/dali.

### Cloning and mutagenesis

The DrT-VTDs were amplified from *Danio rerio* genomic DNA or, in the case of DrT1-VTD2, from *Danio rerio* cDNA (kind gifts from Sigrun Korsching, University of Cologne), Wc-VTD2 was amplified from *Waddlia chondrophila* genomic DNA (kind gift from Carole Kebbi Beghdadi, University of Lausanne), Si-VTD was amplified from *Serendipita indica* cDNA (kind gift from Alga Zuccaro, University of Cologne), and Dm-VTD was amplified from *Drosophila melanogaster* cDNA (kind gift from Mirka Uhlirova, University of Cologne). BPLF1 was amplified from a plasmid (kind gift from Maria Masucci). All amplifications were done by PCR, using Phusion High Fidelity Kit (New England Biolabs). Wc-VTD1, Nf-VTD, Sl-VTD and M48coding regions were obtained by gene synthesis (IDT). The PCR fragments and gBlocks were cloned into pOPIN-S and pOPIN-K vectors[61] using the In-Fusion HD Cloning Kit (Takara Clontech). Point mutations were introduced using the QuikChange Lightning kit (Agilent Technologies).

Constructs for ubiquitin-PA purification (pTXB1-ubiquitin$^{1–75}$) were a kind gift of David Komander (WEHI, Melbourne). SUMO1$^{1–96}$, SUMO3$^{1–91}$ and ISG15$^{79–156}$ were amplified by PCR with an N-terminal 3xFlag tag and cloned into the pTXB1 vector (New England Biolabs) by restriction cloning according to the manufacturers protocol. The Ub$^{I44A}$ activity-based probe was created by introducing the mutation using the QuikChange Lightning kit (Agilent Technologies).

### Protein expression and purification

All VTD candidates including all truncations and mutants as well as BPLF1 and M48 were expressed from the pOPIN-S (in case of the bacterial and eukaryotic proteases) with an N-terminal 6His-Smt3-tag or pOPIN-K vector (in case of the transposon-derived candidates) with an N-terminal 6His-GST-tag. *Escherichia coli* (Strain: Rosetta (DE3) pLysS) were transformed with constructs expressing DUBs and 2–6 l cultures were grown in LB medium at 37 °C until the OD600 of 0.8 was reached. The cultures were cooled down to 18 °C and protein expression was induced by addition of 0.2 mM isopropyl β-d-1-thiogalactopyranoside (IPTG).

The expression of selenomethionine substituted proteins was carried out as described previously by[62]: In brief, the expression cultures were grown in minimal medium until the OD600 of 0.8 was reached. The cultures were cooled down to 18 °C, mixed with feedback inhibition amino acid mix (0.5 g/l final concentration), metal trace elements (0.1% final concentration) and vitamins (0.01% final concentration) and induced with 0.2 mM IPTG. After 16 h, the cultures were harvested by centrifugation at 5000 × g for 15 min. After freeze thaw, the pellets were resuspended in binding buffer (300 mM NaCl, 20 mM TRIS pH 7.5, 20 mM imidazole, 2 mM β-mercaptoethanol) containing DNase and Lysozyme, and lysed by sonication using 10 s pulses with 50 W for a total time of 10 min. Lysates were clarified by centrifugation at 50,000 × g for 1 h at 4 °C and the supernatant was used for affinity purification on HisTrap FF columns (GE Healthcare) according to the manufacturer's instructions. The 6His-Smt3 tag was removed by incubation with SENP1$^{415–644}$; the 6His-GST tag was removed by incubation with 3 C protease. Si-VTD and Sl-VTD were purified including the N-terminal 6His-Smt3 tag. The proteins were simultaneously dialyzed in binding buffer. The liberated affinity-tag and the His-tagged SENP1 and 3 C proteases were removed by a second round of affinity purification with HisTrap FF columns (GE Healthcare). All proteins were purified with a final size exclusion chromatography (HiLoad 16/600 Superdex 75 or 200 pg) in 20 mM TRIS pH 7.5, 150 mM NaCl, 2 mM dithiothreitol (DTT), concentrated using VIVASPIN 20 Columns (Sartorius), flash frozen in liquid nitrogen, and stored at

−80 °C. Protein concentrations were determined using the absorption at 280 nm ($A_{280}$) using the proteins' extinction coefficients derived from their sequences.

## Synthesis of activity-based probes

All activity-based probes used in this study were expressed as C-terminal intein fusion proteins as described previously[63]: In brief, the fusion proteins were affinity purified in buffer A (20 mM HEPES, 50 mM sodium acetate pH 6.5, 75 mM NaCl) from clarified lysates using Chitin Resin (New England Biolabs) following the manufacturer's protocol. On-bead cleavage was performed by incubation with cleavage buffer (buffer A containing 100 mM MesNa (sodium 2-mercaptoethane-sulfonate)) for 24 h at room temperature (RT). The resin was washed extensively with buffer A and the pooled fractions were concentrated and subjected to size exclusion chromatography (HiLoad 16/600 Superdex 75 pg) with buffer A. To synthesize the propargylated probe, 300 µM Ub/Ubl-MesNa were reacted with 600 µM propargylamine hydrochloride (Sigma Aldrich) in buffer A containing 150 mM NaOH for 3 h at RT. Unreacted propargylamine was removed by size exclusion chromatography and the probe was concentrated using VIVASPIN 20 Columns (3 kDa cutoff, Sartorius), flash frozen and stored at −80 °C. The NEDD8-PA was a kind gift from David Pérez Berrocal and Monique Mulder (Department of Cell and Chemical Biology, Leiden University).

## Chain generation

Met1-linked di-ubiquitin was expressed as a linear fusion protein and purified by ion exchange chromatography and size exclusion chromatography. K6-, K11-, K48-, and K63-linked ubiquitin chains were enzymatically assembled using UBE2SΔC (K11), CDC34 (K48), and Ubc13/UBE2V1 (K63) as previously described[64,65]. In brief, ubiquitin chains were generated by incubation of 1 µM E1, 25 µM of the respective E2, and 2 mM ubiquitin in reaction buffer (10 mM ATP, 40 mM TRIS (pH 7.5), 10 mM $MgCl_2$, 1 mM DTT) for 18 h at RT. K6-linked ubiquitin chains were assembled by incubation of 1 µM E1, 25 µM E2 (UbCH7), 25 µM E3 (NleI) and 2 mM wildtype or F4A ubiquitin in reaction buffer. The generated mixture of K6- and K48-linked chains was treated with 10 µM OTUB1. The respective reactions were stopped by 20-fold dilution in 50 mM sodium acetate (pH 4.5) and chains of different lengths were separated by cation exchange using a Resource S column (GE Healthcare). Elution of different chain lengths was achieved with a gradient from 0 to 600 mM NaCl.

## Crystallization

DrT1-VTD2 (selenomethionine substituted), Wc-VTD1 (selenomethionine substituted) and Wc-VTD1-Ub were crystallized using sitting drop vapor diffusion with commercially available sparse matrix screens. 96 well iQ crystallization plates containing 30 µl of the respective screening conditions were mixed with 10 mg/ml protein in the ratios 1:2, 1:1 and 2:1 in 300 nl drops. For DrT1-VTD2 and Wc-VTD1, the initial conditions containing the most promising crystals were optimized by gradually changing the chemical components included in the respective screening condition. 80 µl reservoir solution were pipetted into 48-well MRC plates and sitting drop vapour diffusion was performed by mixing 10 mg/ml protein in the ratios 1:2, 1:1 and 2:1 in drops of 3 µl. Initial DrT1-VTD2 crystals were detected in Proplex D12 (0.1 M Tris pH 8.5, 20% w/v PEG 6000) and several other conditions after 3 days at 20 °C. Optimization was carried out with 3 µl drops (protein/precipitant ratios: 2:1, 1:1 and 1:2) and precipitant solutions varying in pH or PEG 6000 concentration respectively. Optimized crystals were harvested and cryoprotected with reservoir containing 10% glycerol. For unbound Wc-VTD1, initial crystals were detected in JCSG H8 (0.2 M sodium chloride, 0.1 M BisTris pH 5.5, 25% w/v PEG 3350) and several other conditions after 3 days at 20 °C. Optimization was carried out with 3 µl drops (protein/precipitant ratios: 2:1, 1:1 and 1:2) and precipitant solutions varying in sodium chloride or PEG 3350

concentration respectively. Optimized crystals were harvested and cryoprotected with reservoir containing 20% glycerol. 1.2 mM Wc-VTD1 were incubated with 760 µM ubiquitin-PA for 18 h at 4 °C. Unreacted Wc-VTD1 and Ub-PA were removed by size exclusion chromatography and the complex concentrated to a concentration of 7 mg/ml. The covalent Wc-VTD1-Ub crystals were detected in Salt RX B6 (0.1 M BisTris propane pH 7.0, 2.0 M Ammonium citrate tribasic) after 2 days at 20 °C. Crystals were harvested without further optimization and cryoprotected with the addition of 10% glycerol.

## Data collection, phasing, model building, and refinement

Diffraction data for DrT1-VTD2, Wc-VTD1 and the Wc-VTD1-Ub-PA complex was collected at the Deutsches Elektronen-Synchroton (DESY), Hamburg, Germany at beamline P13 at the EMBL outstation[66]. All datasets were processed using XDS[67]. For DrT1-VTD2 and Wc-VTD1, initial phases were determined using selenomethionine SAD experiments and SHELXC/D/E[68], which were driven by HKL2MAP[69]. Both protein structures were subsequently built automatically using the ARP/wARP Web Service[70] or the buccaneer pipeline implemented in CCP4[71]. The Wc-VTD1-Ub-PA complex was solved by molecular replacement using PHASER[72] as implemented in the phenix package[73] with a single molecule of Wc-VTD1 and a full-length model of ubiquitin as search models. For further refinement, necessary restraints for the propargyl moiety were calculated using AceDRG implemented in the CCP4 package[74]. Initial models were refined using iterative cycles of phenix.refine (DrT1-VTD2, Wc-VTD1) or RefMac (Wc-VTD1-Ub-PA) and manually rebuilt using COOT[75]. For structural analysis, the PyMOL (http://www.pymol.org) and ChimeraX Graphics Systems[76] were used.

## AMC assays

Activity assays of DUBs against AMC-labeled substrates were performed using reaction buffer (150 mM NaCl, 20 mM TRIS pH 7.5, 10 mM DTT), 1 µM DUBs (deviating concentrations are indicated in the respective figures and the corresponding legends), 5 µM Ub-AMC (UbiQ-Bio, The Netherlands), 5 µM NEDD8-AMC (Enzo Life Science) or 100 µM zRLRGG-AMC (BACHEM AG, Switzerland). The reaction was performed in black 96-well plates (Corning) at 30 °C and fluorescence was measured using the Infinite F200 Pro plate reader (Tecan) equipped for excitation wavelength of 360 nm and an emission wavelength of 465 nm. The data was collected using Magellan 7.1 software (Tecan). The presented results are means of three independent cleavage assays.

## Activity-based probe assays

DUBs were prediluted to 2× concentration (10 µM) in reaction buffer (20 mM TRIS pH 7.5, 150 mM NaCl and 10 mM DTT) and 1:1 combined with 100 µM Ub-, Ub$^{I44A}$-, SUMO1, SUMO3, ISG15$^{CTD}$ or NEDD8-PA for 18 hours at 4 °C. The reaction was stopped by the addition of 2x Laemmli buffer, resolved by SDS-PAGE, and Coomassie stained.

## Ubiquitin chain cleavage

DUBs were preincubated in 150 mM NaCl, 20 mM TRIS pH 7.5 and 10 mM DTT for 10 min. The cleavage was performed for the indicated time points with 25 nM up to 500 nM DUBs (as indicated in the respective figure legends) and 25 µM di-ubiquitin (K11, K48, K63, M1, K6 synthesized as described above, others purchased from Boston Biochem) at RT. The reactions were stopped with 2x Laemmli buffer, resolved by SDS-PAGE, and Coomassie stained.

## Statistics and reproducibility

All activity-based probes, chain cleavage, ubiquitin-binding, and AMC assays were performed two independent times with similar results. Each AMC assay was additionally performed in triplicates for noise reduction.

**Reporting summary**

Further information on research design is available in the Nature Portfolio Reporting Summary linked to this article.

## Data availability

The data underlying the findings of this study are available in this article and its Supplementary Information or are available from the corresponding author upon reasonable request. The X-ray structures of DrT1-VTD1 and Wc-VTD in its apo and ubiquitin-bound form have been deposited at the PDB database under the accession numbers 8ADD, 8ADC and 8ADB, respectively. The X-ray structures of M48 and AvrPphB are publicly available at the PDB database under the accession numbers 2J7Q and 1UKF, respectively. Source data are provided with this paper.

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

## Acknowledgements

We thank Christiane Horst and Claudia Poschner for expert technical assistance. We are also grateful to David Pérez Berrocal, Monique Mulder, Sigrun Korsching, Mirka Uhlirova, Maria Masucci and Alga Zuccaro for reagents and constructs. The synchrotron data was collected at beamline P13 operated by EMBL Hamburg at the PETRA III storage ring (DESY, Hamburg, Germany). We would like to thank David von Stetten, Michael Agthe, Gleb Bourenkov, and Isabel Bento for the assistance in using the beamline. Deubiquitinase research in the lab of K.H. is supported by DFG grant HO 3783/3-1. Crystals were grown using equipment of the Cologne Crystallization facility (C$_2$f), which is supported by DFG grant INST 216/949-1 FUGG.

## Author contributions

I.E. performed most biochemical experiments and crystallization, E.A. solved the X-ray structures, T.H. contributed to the experiments, U.B. supervised the crystallography, K.H. initiated and supervised the study and contributed bioinformatical analyses. All authors contributed to data analysis and the writing of the manuscript.

## Funding

## Competing interests

The authors declare no competing interests.
