## [Peer Review File · Nature Communications]

REVIEWER COMMENTS

Reviewer #1 (Remarks to the Author):

The manuscript by Erven et al. reports on the identification and functional characterization of a new family of deubiquitinase expressed in eukaryotes and procaryotes that, based on their similarity with previously described DUBs encoded in the N-terminal domain of the large tegument protein of herpesviruses, are collectively named as viral tegument-like DUB (VTD). Using bioinformatics, structural and functional analysis they propose a scenario where the VTD family may have evolved from a common non-DUB protease precursor expressed in bacteria, acquiring distinctive structural and functional features, such as preference for different ubiquitin-like or poly-ubiquitin linkages, though evolution in different taxa. The paper addresses an interesting aspect of protein structure-function relationship and its significance for protein function evolution. The data are of high quality and clearly presented, making them accessible also to a broad audience of non-specialists.

Specific comments

1. While not strictly essential in the context of the paper data comparing the enzymatic activity of the newly discovered VTDs with that of the founding members of this enzyme family, such as HSV1 UL36 or any other herpesvirus homolog, would provide a valuable reference in terms of structure-function relationship and possible biological activity of the enzymes. For example, data on ubiquitin-AMC and NEDD8-AMC cleavage by a reference VTD could be easily added in Fig 2 for comparison.
2. In the discussion the authors convincingly argue that the VTD family may have evolved from a Yop-T-like precursor gene. The product of this gene are proteases but lacks DUB activity. Is it possible that at least some members of the VTD family may have maintained enzymatic activity against other substrates than ubiquitin and UbLs? Does the structure analysis offer any clue on the possible function/specificity beyond the ubiquitin system?

Reviewer #2 (Remarks to the Author):

(I was asked to review the bioinformatics part of the paper)

The bioinformatics component of this manuscript is poor. The authors describe their methods poorly by citing their own past papers with methods that are also poorly described. A good method has clear objectives. In this case, it should be specified whether the objective is to detect homology or

detect similarity in specific motifs. After defining the objective, one needs to specify and justify a criterion. For example, two genes are considered homologous above this criterion. There is no such criterion presented, either in this paper or in other papers by the authors that have been cited to support the method in this paper. One may infer that the authors were aiming to identify shared motifs and that hopefully such motifs will be indicative of homology. However, this is not explicit.

From a set of aligned sequences, one can generate a position weight matrix that does not take into consideration of site dependence, or one can use a Markov model of order 2, order 3, etc., to model site dependence. One can then test whether a Markov model of order i is minimal and sufficient, i.e., order $i+1$ does significantly improve the model. The manuscript has none of such information.

MAFFT is for global sequence alignment. As DUB sequences are highly diverged, it does not seem the right approach to start with a global MAFFT alignment. Using Gibbs sampler to search for local similarities would seem to be more appropriate.

The manuscript also contains unsubstantiated or contradictory statements. For example, the authors stated that cysteine-DUBs are categorized in six different classes based on sequence and structural similarity, but have never mentioned what sequence or structural similarities are shared among these cysteine proteases and how they are used to classify the cysteine-DUBs into six different classes.

They stated that most bacterial DUBs prefer K63-linked chains or have no strong linkage preferences, then immediately stated that the bacterial LotA is K6-specific.

They stated that “there are sequence similarities between different DUB classes which argue for a common evolutionary ancestry”. They cited two papers of their own, but these two papers did not provide reasonable evidence that different DUB classes share a common ancestor. The authors should point out exactly which paper using what evidence established the coancestry claim.

Reviewer #3 (Remarks to the Author):

The manuscript by Erven et al. reports on the bioinformatic discovery of large groups of proteins with VTD (viral tegument-like DUB) domains which encode active deubiquitinases of varying Ub/Ubl- and surprisingly exquisite Ub chain linkage-specificities. DUBs with this domain have previously only been known to exist in genomes of herpesviruses, but are here shown to also exist in several distinct subgroups and in a broad number of organisms. While no VTDs were identified in humans or higher eukaryotes (*Drosophila* and nematodes are featured), this nevertheless represents a very important discovery as it elevates the VTD DUB family from their viral niche to being the 8th class of deubiquitinases.

The manuscript first describes the bioinformatic search strategy which through iterative processes uncovered several distinct groups of VTDs of which representative examples were selected for recombinant expression and biochemical characterization. A total of 8 VTDs from 7 organisms were selected and studied through Ub/Ubl probe binding assays, Ub/Ubl fluorogenic substrate cleavage assays and gel-based ubiquitin chain cleavage assays. Moreover, the structures of two VTD DUBs (from zebrafish and from the pathogenic *Waddlia* bacterial family), including one structure in complex with monoubiquitin, are presented. While the overall structures are generally similar to the previously known M48 herpesvirus VTD DUB, they differ in key regions which help to rationalize their diverse catalytic activities. Most notably, VTDs of various Ub chain linkage specificities were found which stresses the conceptual similarity of VTDs to the OTU family of DUBs which through customization of a core domain was also evolved to facilitate the specific processing of Ub chains.

All in all, this is a highly relevant and surprising discovery that will find the interest of the Ubiquitin community. It provides a coherent framework for the likely evolutionary path and relationship of VTDs with other hydrolases. The strength of the manuscript is that a rather large number of VTDs from different organisms are sampled which emphasizes the catalytic diversity in this family. However, this in turn means that the study of the VTDs is mainly descriptive and important aspects like the chain linkage specificity are not studied mechanistically. I regard the conservation and diverse cleavage specificities as sufficient indication of biological relevance (which I am sure will be investigated following publication of this report from more biological groups), but in lieu of experimental evidence for expression or physiological roles the mechanistic dissection of the identified activities should be completed before publication can be supported for the chosen journal. I recommend the authors to consider improving it along the following suggested lines.

Main points:

- I understand why the authors chose the current title, but still suggest rephrasing it together with key parts of the abstract and main manuscript: The family of VTDs is not “new” (see reference 23 from 2007, describing VTDs as a new class of active DUBs), and instead it should be highlighted that VTDs can be found in a broad range of organisms. Despite not being new, this is the main and significant finding that should be stated without oversimplification. It may also make sense to use the words family / class consistently. Moreover, the long part “spreading and diversifying through

transposon and horizontal gene transfer” is rather speculative as there is no experimental evidence provided, especially not for the horizontal gene transfer.

- In Figure 3, the structure of the zebrafish VTD is shown which is highly similar to the herpesvirus M48 DUB except for the highlighted loop. However, the biochemical characterization remains inconclusive as to whether both DUBs utilize a similar mechanism for Ub chain selection. This should be experimentally clarified, e.g. by conducting the I44A Ub probe assay also for the M48 DUB. Moreover, the authors could carry out a triple mutation (from the mutations shown in Figure 3i) or any other suitable side-by-side comparison of DrT1-VTD with M48. Since mutation of D376 and E377 are sufficient to abrogate cleavage and the I44 patch does not seem critical for Ub recognition, do VTDs cleave Ub C-terminal substrate (e.g. the LRLRGG-AMC substrate as shown by the authors for zUFSP)?

- In Figure 5, the authors present the structure of strictly K6-specific VTD in complex with mono ubiquitin, leaving the mechanism for linkage specificity open. I recognize that attempts to crystallize the DUB with a diUbiquitin substrate were unsuccessful, but in the absence of further structural data the mechanism should be tested experimentally. This could be accomplished by using Ubiquitin chains with hydrophobic patches mutated (for K6, the Phe4 patch was critical for other DUBs and due to the position of Lys6 may be generally important for this chain type). Moreover, the authors speculate about an S1' site and a role of the alpha9 helix which should be tested by mutation. Guessing S1' sites was previously successful in a similar case (see Pruneda et al Mol Cell 2016).

- While not strictly required from my point of view, the manuscript would benefit from the inclusion of any (even if remote) evidence for the expression of the studied VTDs to confirm (or make appear likely) their existence on the protein level.

Minor points:

- The number of independently performed experiments should be given in the captions or if applicable in a general statement (at least 2 per data element) and uncropped versions of the gels should be included as supplementary information.

- Line 56 / 421: I agree that there is no truly K6-specific DUB known in eukaryotes, but USP30 as a K6-selective DUB and OTUD3 as a K6/K11-dual specific DUB should be mentioned and referenced.

- Line 71: The authors are encouraged to include a citation of Pruneda et al Mol Cell 2016 where activities of the bacterial CE clan proteases are dissected.

- Figure 1 (lines 116 following): I think the readers would benefit from a graphical representation of the iterative search procedure and the resulting discovered groups of VTDs. Moreover, the total length of the chosen proteins could be added to panel b.

- Figure 2 and following: Please quantify the AMC cleavage activities for comparison (e.g. with catalytic efficiencies).

- Line 185: Please explicitly mention the previously observed specificities for herpesvirus VTDs for easy comparison.

- Line 193: Please number secondary structure elements.
- Line 196: Generally unstructured means that there is no sufficient density to build the amino acids, whereas the authors here use it meaning “having no secondary structure”. Please rephrase to avoid confusion.
- The discussion contains various rather speculative paragraphs, and I would ask the authors to substantiate their analysis through the inclusion of the relevant data (e.g. the mentioned dendrogram, line 352). Moreover, a graphical summary of the evolutionary pathways and the relationship of the various family may be beneficial.
- Line 597: I assume this means VTD1 not VTD2?
- Line 617: Please add units for B factors and RMS data.
- Figure 2f: Please compare data measured at the same concentration (currently 5 nM vs 1 nM)
- Figure 2j,k,l: Please perform additional cleavage assays with lower enzyme concentration of the cleaved chains. This is important as there is complete turnover already at the first time point and further specificity could be masked under those conditions.
- I would expect both ubiquitin and the Ubls to be fully conserved in the investigated species (where appropriate, e.g. not for Waddlia), but I would ask the authors to check. It would not be necessary to repeat assays with customized reagents, but important to comment.
- Figure 5 (and others): Please increase the labeling of elements in the structure figures to increase accessibility.
- Figure 5d: Please also show Pro383 as this is discussed in the text.
- Figure 6j,kl and Figure 7d: What is the reason for the seemingly dirty Lys63 chains (and uneven loading on Figure 7d)? If possible, this should be repeated with clean chains (or if chains are clean without boiling of the samples prior loading on the gel).

Reviewer #1 (Remarks to the Author):

The manuscript by Erven et al. reports on the identification and functional characterization of a new family of deubiquitinase expressed in eukaryotes and procaryotes that, based on their similarity with previously described DUBs encoded in the N-terminal domain of the large tegument protein of herpesviruses, are collectively named as viral tegument-like DUB (VTD). Using bioinformatics, structural and functional analysis they propose a scenario where the VTD family may have evolved from a common non-DUB protease precursor expressed in bacteria, acquiring distinctive structural and functional features, such as preference for different ubiquitin-like or poly-ubiquitin linkages, though evolution in different taxa. The paper addresses an interesting aspect of protein structure-function relationship and its significance for protein function evolution. The data are of high quality and clearly presented, making them accessible also to a broad audience of non-specialists.

Specific comments

1. While not strictly essential in the context of the paper data comparing the enzymatic activity of the newly discovered VTDs with that of the founding members of this enzyme family, such as HSV1 UL36 or any other herpesvirus homolog, would provide a valuable reference in terms of structure-function relationship and possible biological activity of the enzymes. For example, data on ubiquitin-AMC and NEDD8-AMC cleavage by a reference VTD could be easily added in Fig 2 for comparison.

We agree with Reviewer #1 and compared the cleavage of ubiquitin- and Nedd8-AMC by the helitron-encoded VTDs to the EBV tegument DUB BPLF1. The data was added to Fig2 d-i and are described on page 7. On a related note, we also compared the mechanism of Ub-binding by the helitron-VTD to the M48 tegument DUB from murine CMV, as requested by Reviewer #3. These data were added to Fig. 3j.

2. In the discussion the authors convincingly argue that the VTD family may have evolved from a YopT-like precursor gene. The product of this gene are proteases but lacks DUB activity. Is it possibility that at least some members of the VTD family may have maintained enzymatic activity against other substrates than ubiquitin and UbLs? Does the structure analysis offer any clue on the possible function/specificity beyond the ubiquitin system?

An interesting question, which is difficult to answer. The YopT/HopM family includes several members with fundamentally different substrates and cleavage site preferences. Thus, there is no 'generic YopT-assay', which could be used for testing if VTDs have retained a substrate-specific protease activity. The structures of known VTDs show a conservation of features required for ubiquitin recognition, consistent with the idea that ubiquitin and/or Nedd8 removal is their main function. However, it still remains possible that other activities exist in these proteins, which could only be uncovered through an unbiased screen in cells from their respective organisms.

Reviewer #2 (Remarks to the Author):

(I was asked to review the bioinformatics part of the paper)

The bioinformatics component of this manuscript is poor.

We are sorry to hear that our bioinformatical reasoning wasn't made sufficiently clear. We have now made efforts to better document the steps involved and the significance criteria applied. It should be

noted, however, that in the present manuscript our bioinformatical claims were also validated experimentally.

The authors describe their methods poorly by citing their own past papers with methods that are also poorly described. A good method has clear objectives. In this case, it should be specified whether the objective is to detect homology or detect similarity in specific motifs. After defining the objective, one needs to specify and justify a criterion. For example, two genes are considered homologous above this criterion. There is no such criterion presented, either in this paper or in other papers by the authors that have been cited to support the method in this paper. One may infer that the authors were aiming to identify shared motifs and that hopefully such motifs will be indicative of homology. However, this is not explicit.

We used the L-INS-I algorithm of the MAFFT package to align the deubiquitinase domains of herpesviral tegument-DUBs and created from this alignment a generalized profile, which was subsequently used for the first round of database searches. This method was co-developed in 1996 by author KH, making it necessary to cite our own past paper. We are not aware of having cited another method-centric paper of ours to support the bioinformatics section. The aim of this database search was to identify other sequences with significant similarity to the query profile. To make this clearer, we have added an additional explanation to the first paragraph of the results section and added a significance criterion to the Methods section describing bioinformatics.

From a set of aligned sequences, one can generate a position weight matrix that does not take into consideration of site dependence, or one can use a Markov model of order 2, order 3, etc., to model site dependence. One can then test whether a Markov model of order i is minimal and sufficient, i.e., order $i+1$ does significantly improve the model. The manuscript has none of such information.

The 'generalized profile' (GP) method used by us is a position-specific weight matrix approach, which also includes position-specific and asymmetric gap penalties. The method is described in detail in the 1996 Computational Chemistry paper cited by us. This publication also documents the equivalence of the GP method and the widely-used (1st order) profile-HMM approaches (e.g. used by HMMER). We feel that a description of the inner workings of the search method is outside the scope of the current manuscript.

MAFFT is for global sequence alignment. As DUB sequences are highly diverged, it does not seem the right approach to start with a global MAFFT alignment. Using Gibbs sampler to search for local similarities would seem to be more appropriate.

In the present case, the global alignment properties of MAFFT do not pose a problem. In the current manuscript, we use MAFFT for aligning catalytic domains rather than entire proteins. Like other deubiquitinase domains, the VTD domain is a true 'homology domain' characterized by a common fold and well-defined domain boundaries, which are detectable by a sharp drop in sequence conservation outside the domain. We have added a clarifying sentence to the bioinformatics section of the result.

The manuscript also contains unsubstantiated or contradictory statements. For example, the authors stated that cysteine-DUBs are categorized in six different classes based on sequence and structural similarity, but have never mentioned what sequence or structural similarities are shared among these cysteine proteases and how they are used to classify the cysteine-DUBs into six different classes.

The classification of cysteine-DUBs into six different classes is not of our making; it is mainly based on sequence- and structure-relationship and has evolved among ubiquitin researchers over the past years. The classification is extensively documented in the review literature, in particular references #11 (Clague et al. 2019) and #12 (Pruneda et al 2019). We have added these references when mentioning the classification system within the introduction section (page 3).

They stated that most bacterial DUBs prefer K63-linked chains or have no strong linkage preferences, then then immediately stated that the bacterial LotA is K6-specific.

We apologize for making this look like a contradiction. Most bacterial DUBs are unspecific or prefer K63 chains. The two big exceptions are LotA and RavD, both from Legionella. We have added a remark to the introduction section (page 3) emphasizing the exceptional nature of these two Legionella activities.

They stated that “there are sequence similarities between different DUB classes which argue for a common evolutionary ancestry”. They cited two papers of their own, but these two papers did not provide reasonable evidence that different DUB classes share a common ancestor. The authors should point out exactly which paper using what evidence established the coancestry claim.

We admit that there is no proof for a common evolutionary ancestry of cysteine-DUBs and have therefore removed this claim from the manuscript. We had noticed a conspicuous relationship between DUB classes while doing the all-against-all comparison of cysteine protease families for our 2020 LSA paper (reference #18), but we did not provide a formal proof for co-ancestry. We have modified the introduction section (page 4) and removed the claim for common ancestry. The 2nd claim (sequence relationship between different DUB families) was left in place, since this is documented by several examples in the LSA paper (reference #18) and an older paper (reference #19) establishing similarity between the USP, UCH and Josephin families.

Reviewer #3 (Remarks to the Author):

The manuscript by Erven et al. reports on the bioinformatic discovery of large groups of proteins with VTD (viral tegument-like DUB) domains which encode active deubiquitinases of varying Ub/Ubl- and surprisingly exquisite Ub chain linkage-specificities. DUBs with this domain have previously only been known to exist in genomes of herpesviruses, but are here shown to also exist in several distinct subgroups and in a broad number of organisms. While no VTDs were identified in humans or higher eukaryotes (Drosophila and nematodes are featured), this nevertheless represents a very important discovery as it elevates the VTD DUB family from their viral niche to being the 8th class of deubiquitinases.

The manuscript first describes the bioinformatic search strategy which through iterative processes uncovered several distinct groups of VTDs of which representative examples were selected for recombinant expression and biochemical characterization. A total of 8 VTDs from 7 organisms were selected and studied through Ub/Ubl probe binding assays, Ub/Ubl fluorogenic substrate cleavage assays and gel-based ubiquitin chain cleavage assays. Moreover, the structures of two VTD DUBs (from zebrafish and from the pathogenic Waddlia bacterial family), including one structure in complex with monoubiquitin, are presented. While the overall structures are generally similar to the

previously known M48 herpesvirus VTD DUB, they differ in key regions which help to rationalize their diverse catalytic activities. Most notably, VTDs of various Ub chain linkage specificities were found which stresses the conceptual similarity of VTDs to the OTU family of DUBs which through customization of a core domain was also evolved to facilitate the specific processing of Ub chains.

All in all, this is a highly relevant and surprising discovery that will find the interest of the Ubiquitin community. It provides a coherent framework for the likely evolutionary path and relationship of VTDs with other hydrolases. The strength of the manuscript is that a rather large number of VTDs from different organisms are sampled which emphasizes the catalytic diversity in this family. However, this in turn means that the study of the VTDs is mainly descriptive and important aspects like the chain linkage specificity are not studied mechanistically. I regard the conservation and diverse cleavage specificities as sufficient indication of biological relevance (which I am sure will be investigated following publication of this report from more biological groups), but in lieu of experimental evidence for expression or physiological roles the mechanistic dissection of the identified activities should be completed before publication can be supported for the chosen journal. I recommend the authors to consider improving it along the following suggested lines.

Main points:

- I understand why the authors chose the current title, but still suggest rephrasing it together with key parts of the abstract and main manuscript: The family of VTDs is not “new” (see reference 23 from 2007, describing VTDs as a new class of active DUBs), and instead it should be highlighted that VTDs can be found in a broad range of organisms. Despite not being new, this is the main and significant finding that should be stated without oversimplification. It may also make sense to use the words family / class consistently. Moreover, the long part “spreading and diversifying through transposon and horizontal gene transfer” is rather speculative as there is no experimental evidence provided, especially not for the horizontal gene transfer.

As requested, we have changed the title to „VTD – a widely distributed family of eukaryotic and bacterial deubiquitinases related to herpesviral large tegument proteins” . We have also slightly changed the abstract to better conform to the changed title.

- In Figure 3, the structure of the zebrafish VTD is shown which is highly similar to the herpesvirus M48 DUB except for the highlighted loop. However, the biochemical characterization remains inconclusive as to whether both DUBs utilize a similar mechanism for Ub chain selection. This should be experimentally clarified, e.g. by conducting the I44A Ub probe assay also for the M48 DUB. Moreover, the authors could carry out a triple mutation (from the mutations shown in Figure 3i) or any other suitable side-by-side comparison of DrT1-VTD with M48.

We agree with reviewer #3 and added several experiments to better understand the mechanism and compare it to M48. We generated a quadruple mutant (F398A, P405A, A406G and L407A) of DrT1-VTD2, which completely abrogated the activity against di-ubiquitin, thereby confirming that binding of the Ile44-patch by the flexible loop is crucial for DUB activity (new Fig 3j). We previously showed that binding to Ile44 itself is not necessary for Ub-PA reactivity in case of DrT1-VTD2 (old Fig 3j/new Fig 3h). In addition, the published crystal structure of the M48~Ub complex (PDB: 2J7Q) shows intensive contacts of M48 to ubiquitin His68 and Val70, but not to Ile44 itself. We therefore added M48 to our I44A probe assay and could confirm that direct binding to Ile44 is dispensable for M48 as well (new fig 3h). The text (page 9ff) was changed accordingly to document the newly added results.

Since mutation of D376 and E377 are sufficient to abrogate cleavage and the I44 patch does not seem critical for Ub recognition, do VTDs cleavage Ub C-terminal substrate (e.g. the LRLGG-AMC substrate as shown by the authors for zUFSP)?

DrT1-VTD2 is able to cleave the RLRGG-AMC peptide substrate as expected by reviewer #3. The mutants show the same phenotype as for Ub-AMC cleavage, confirming the crucial stabilization of Arg72/74. These data were added to Supplementary Figure 3h. Please note that we do **not** claim that the entire I44 patch is dispensable for cleavage - this only applies to the Ile44 residue itself.

- In Figure 5, the authors present the structure of strictly K6-specific VTD in complex with mono ubiquitin, leaving the mechanism for linkage specificity open. I recognize that attempts to crystallize the DUB with a diUbiquitin substrate were unsuccessful, but in the absence of further structural data the mechanism should be tested experimentally. This could be accomplished by using Ubiquitin chains with hydrophobic patches mutated (for K6, the Phe4 patch was critical for other DUBs and due to the position of Lys6 may be generally important for this chain type). Moreover, the authors speculate about an S1' site and a role of the alpha9 helix which should be tested by mutation. Guessing S1' sites was previously successfully in a similar case (see Pruneda et al Mol Cell 2016).

We addressed these issues by performing two additional experimental series. First, we followed the reviewer's suggestion and tested the importance of ubiquitin's Phe4 patch, which was recently shown to be important for K6-specificity of Legionella LotA. Unlike LotA, Wc-VTD1 could cleave K6-linked di-ubiquitin harboring a F4A mutant, indicating that a different mechanism is used to achieve the specificity. The data was added to Supplementary Figure 4c.

We also tried to 'guess' residues that might be involved in S1'-recognition and tested them for their importance. We focused on surface-exposed residues of helices $\alpha 1$ and $\alpha 9$, mutagenized six of them and found that both regions make crucial contributions to S1'-recognition (new Figure 5i). The text describing these results has been added on page 11 of the revised manuscript.

- While not strictly required from my point of view, the manuscript would benefit from the inclusion of any (even if remote) evidence for the expression of the studied VTDs to confirm (or make appear likely) their existence on the protein level.

VTD proteins are not encoded by the genomes of human, mouse and budding yeast – the three organisms for which the most extensive proteomics datasets are available. There are limited proteomics datasets available for Drosophila and C. elegans, but they do not include the VTD proteins from these organisms – most likely due to their highly-restricted expression in the male germ line (Ref #35 Leader et al, Ref #58 Ebbing et al). The gene deletion phenotype of drosophila "male sterile (3)76Ca" is indicative of its physiological relevance, but does not constitute a proof at the proteome level. However, the corresponding CG14101 protein was found several times as a hit in co-IP experiments tabulated in BioGrid, suggesting that the protein is indeed being formed. Even one of the bacterial VTDs – the divergent Wc-VTD2/wcw_1327 – was previously reported as an immunogenic protein (Ref #44, Kebbi-Beghdadi et al). We have added this citation to the revised manuscript.

Minor points:

- The number of independently performed experiments should be given in the captions or if

applicable in a general statement (at least 2 per data element) and uncropped versions of the gels should be included as supplementary information.

A general statement was added to the methods section. Uncropped gels as well as AMC source data will be provided as a source data file according to the journal policy.

- Line 56 / 421: I agree that there is no truly K6-specific DUB known in eukaryotes, but USP30 as a K6-selective DUB and OTUD3 as a K6/K11-dual specific DUB should be mentioned and referenced.

OTUD3 was already mentioned and referenced in lines 422f. We have now additionally mentioned and referenced USP30 as further eukaryotic K6-prefering DUB (page 15)

- Line 71: The authors are encouraged to include a citation of Pruneda et al Mol Cell 2016 where activities of the bacterial CE clan proteases are dissected.

The Citation was added.

- Figure 1 (lines 116 following): I think the readers would benefit from a graphical representation of the iterative search procedure and the resulting discovered groups of VTDs. Moreover, the total length of the chosen proteins could be added to panel b.

The total length of the proteins was added to figure 1b. We have also created a graphical representation of the search procedure and added it as new supplementary figure 1.

- Figure 2 and following: Please quantify the AMC cleavage activities for comparison (e.g. with catalytic efficiencies).

For comparison to the established viral tegument protease BPLF1 (from Epstein-Barr virus), we have added the activity data of the viral enzyme at the same enzyme concentration (Fig 2d-i, Supplementary Fig 3a). It is clearly visible that DrT1-VTD1 and DrT1-VTD2 are similarly efficient as BPLF1 against Ub-AMC, but are much more efficient against NEDD8-AMC than the viral enzyme. By contrast, DrT2-VTD is much less active than BPLF1 (or the DrT1-enzymes) in cleaving either AMC substrate.

- Line 185: Please explicitly mention the previously observed specificities for herpesvirus VTDs for easy comparison.

The requested information was added

- Line 193: Please number secondary structure elements.

Secondary structure elements are now numbered in Figure 3a and Figure 5b.

- Line 196: Generally unstructured means that there is no sufficient density to build the amino acids, whereas the authors here use it meaning "having no secondary structure". Please rephrase to avoid confusion.

Line 196 was rephrased.

- The discussion contains various rather speculative paragraphs, and I would ask the authors to substantiate their analysis through the inclusion of the relevant data (e.g. the mentioned dendrogram, line 352). Moreover, a graphical summary of the evolutionary pathways and the relationship of the various family may be beneficial.

We have substantiated our claims by adding the requested dendrogram (new Supplementary figure 5) and by providing a graphical summary of the most likely evolutionary events (new Supplementary figure 6).

- Line 597: I assume this means VTD1 not VTD2?

Yes. The typo was corrected.

- Line 617: Please add units for B factors and RMS data.

Missing units were added.

- Figure 2f: Please compare data measured at the same concentration (currently 5 nM vs 1 nM)

We agree with reviewer #3 that the measurement should have been performed with same concentrations. We initially measured Nedd8-AMC cleavage (Fig. 2g) using 5 nM DrT1-VTD2. However, the measurement was out of range already after ~ 5min. Therefore, we lowered concentration to 1nM to stay within the dynamic range. However, to comply with the request, we have now added the 5nM data to Supplementary Figure 3a.

- Figure 2j,k,l: Please perform additional cleavage assays with lower enzyme concentration of the cleaved chains. This is important as there is complete turnover already at the first time point and further specificity could be masked under those conditions.

The assays were repeated using a lower DUB concentration and confirmed the preference of K48- and K63-linked chains. The data were added to new Supplementary Figures 2a-c.

- I would expect both ubiquitin and the Ubls to be fully conserved in the investigated species (where appropriate, e.g. not for Waddlia), but I would ask the authors to check. It would not be necessary to repeat assays with customized reagents, but important to comment.

Ubiquitin is highly conserved among all eukaryotic species tested here, with a maximum of 3aa changes, far from the cleavage/recognition site. We have added a sentence justifying our experiments with human ubiquitin on page 6.

- Figure 5 (and others): Please increase the labeling of elements in the structure figures to increase accessibility.

The labelling was increased in Figures 3c, 3d and 5d to a consistent font size of 8 pt.

- Figure 5d: Please also show Pro383 as this is discussed in the text.

Pro383 is now depicted in Figure 5d.

- Figure 6j,kl and Figure 7d: What is the reason for the seemingly dirty Lys63 chains (and uneven loading on Figure 7d)? If possible, this should be repeated with clean chains (or if chains are clean without boiling of the samples prior loading on the gel).

We thank reviewer #3 for the advice, but additional bands visible in the di-K63 chains are not due to contaminations (confirmed by intact MS analysis). We are aware of that smearing problem, but did not find a good way to solve it. Not boiling the samples prevents smearing in all chain types, except for K63 linked chains. We generated several batches of K63-linked di-ubiquitin and always observe the same behavior. However, we don't think that the smear is a major issue, since we didn't quantify the cleavage based on these gels.

Panel 7d was repeated with equal loading and the panel was replaced.

REVIEWERS' COMMENTS

Reviewer #3 (Remarks to the Author):

The authors have resubmitted an improved version of the manuscript having taken up advice from all reviewers. The additional data, the clearer presentation of the bioinformatics procedure and the more nuanced title/claims are convincing and I support publication of the manuscript in its current form which will be of broad interest to the Ubiquitin and protein/structure evolution communities.

While not required, the authors are encouraged to consider two rather easy additions to make their additional data in Fig. 5i (with the S1' mutants) more accessible:

1. A structure figure which shows the location of these mutated residues would be beneficial.
2. If possible, a rather quick Ub-AMC with any one of those mutants that are inactive in the diUb assay would unequivocally demonstrate that it is an S1' site mutant (as opposed to a mutation that kills the overall activity).

Reviewer #3 (Remarks to the Author):

The authors have resubmitted an improved version of the manuscript having taken up advice from all reviewers. The additional data, the clearer presentation of the bioinformatics procedure and the more nuanced title/claims are convincing and I support publication of the manuscript in its current form which will be of broad interest to the Ubiquitin and protein/structure evolution communities.

While not required, the authors are encouraged to consider two rather easy additions to make their additional data in Fig. 5i (with the S1' mutants) more accessible:

1. A structure figure which shows the location of these mutated residues would be beneficial.

The requested figure was added as Supplementary Fig 4d.

2. If possible, a rather quick Ub-AMC with any one of those mutants that are inactive in the diUb assay would unequivocally demonstrate that it is an S1' site mutant (as opposed to a mutation that kills the overall activity).

Unfortunately, this experiment will not allow meaningful insights, since the wildtype enzyme is already unable to cleave Ub-AMC (see Figure 4c). Thus, any deleterious effects of the S1' site mutants cannot be detected.